# Oncogene-induced senescence in hematopoietic progenitors features myeloid restricted hematopoiesis, chronic inflammation and histiocytosis

Riccardo Biavasco[1,9], Emanuele Lettera[1,2,8,9], Kety Giannetti[1], Diego Gilioli[1,2], Stefano Beretta[1,3], Anastasia Conti[1], Serena Scala[1], Daniela Cesana [1], Pierangela Gallina[1], Margherita Norelli[4], Luca Basso-Ricci[1], Attilio Bondanza[4], Giulio Cavalli[2,5], Maurilio Ponzoni[2,6], Lorenzo Dagna [2,5], Claudio Doglioni [2,7], Alessandro Aiuti [1,2], Ivan Merelli[3], Raffaella Di Micco [1,10 ✉] & Eugenio Montini [1,10 ✉]

Activating mutations in the BRAF-MAPK pathway have been reported in histiocytoses, hematological inflammatory neoplasms characterized by multi-organ dissemination of pro-inflammatory myeloid cells. Here, we generate a humanized mouse model of transplantation of human hematopoietic stem and progenitor cells (HSPCs) expressing the activated form of BRAF (*BRAF^V600E*). All mice transplanted with *BRAF^V600E*-expressing HSPCs succumb to bone marrow failure, displaying myeloid-restricted hematopoiesis and multi-organ dissemination of aberrant mononuclear phagocytes. At the basis of this aggressive phenotype, we uncover the engagement of a senescence program, characterized by DNA damage response activation and a senescence-associated secretory phenotype, which affects also non-mutated bystander cells. Mechanistically, we identify TNFα as a key determinant of paracrine senescence and myeloid-restricted hematopoiesis and show that its inhibition dampens inflammation, delays disease onset and rescues hematopoietic defects in bystander cells. Our work establishes that senescence in the human hematopoietic system links oncogene-activation to the systemic inflammation observed in histiocytic neoplasms.

[1] San Raffaele Telethon Institute for Gene Therapy, IRCCS San Raffaele Scientific Institute, Milan, Italy. [2] Vita-Salute San Raffaele University, Milan, Italy. [3] National Research Council, Institute for Biomedical Technologies, Milan, Italy. [4] Innovative Immunotherapies Unit, IRCCS San Raffaele Scientific Institute, Milan, Italy. [5] Unit of Immunology, Rheumatology, Allergy and Rare Diseases (UnIRAR), IRCCS San Raffaele Scientific Institute, Milan, Italy. [6] Pathology and Lymphoid Malignancies Unit, IRCCS San Raffaele Scientific Institute, Milan, Italy. [7] Unit of Pathology, IRCCS San Raffaele Scientific Institute, Milan, Italy. [8] Present address: Human Oncology and Pathogenesis Program, Department of Radiation Oncology, Memorial Sloan Kettering Cancer Center, New York, US. [9] These authors contributed equally: Riccardo Biavasco, Emanuele Lettera. [10] These authors jointly supervised: Raffaella Di Micco and Eugenio Montini. ✉email: dimicco.raffaella@hsr.it; montini.eugenio@hsr.it

Constitutive activation of MAPK pathway, mostly by mutations in proto-oncogenes *RAS* and *RAF*, is the best-characterized inducer of Oncogene-Induced Senescence (OIS)[1]. OIS prevents cancer progression by halting the proliferation of oncogene-expressing cells, thus dampening the risk of malignant transformation[2]. Proliferation arrest is associated to the accumulation of DNA damage, activation of cell-cycle inhibitors p16[INK4A] and p53[3], widespread chromatin and metabolic remodeling, and sustained secretion of pro-inflammatory cytokines and chemokines, collectively named senescence-associated secretory phenotype (SASP). SASP is crucial for the self-maintenance of OIS and to alert the immune system that damaged cells are present; however, if persistent, it may generate a chronic, detrimental inflammatory milieu, as also reported in age-related diseases[4,5]. Although several studies investigated the impact of oncogene activation and cell senescence in epithelial cells and fibroblasts, the effects of senescence triggers on hematopoietic stem and progenitor cells (HSPCs) and their functional consequences on hematopoietic output remain undetermined. MAPK activating mutations in hematopoietic cells, most

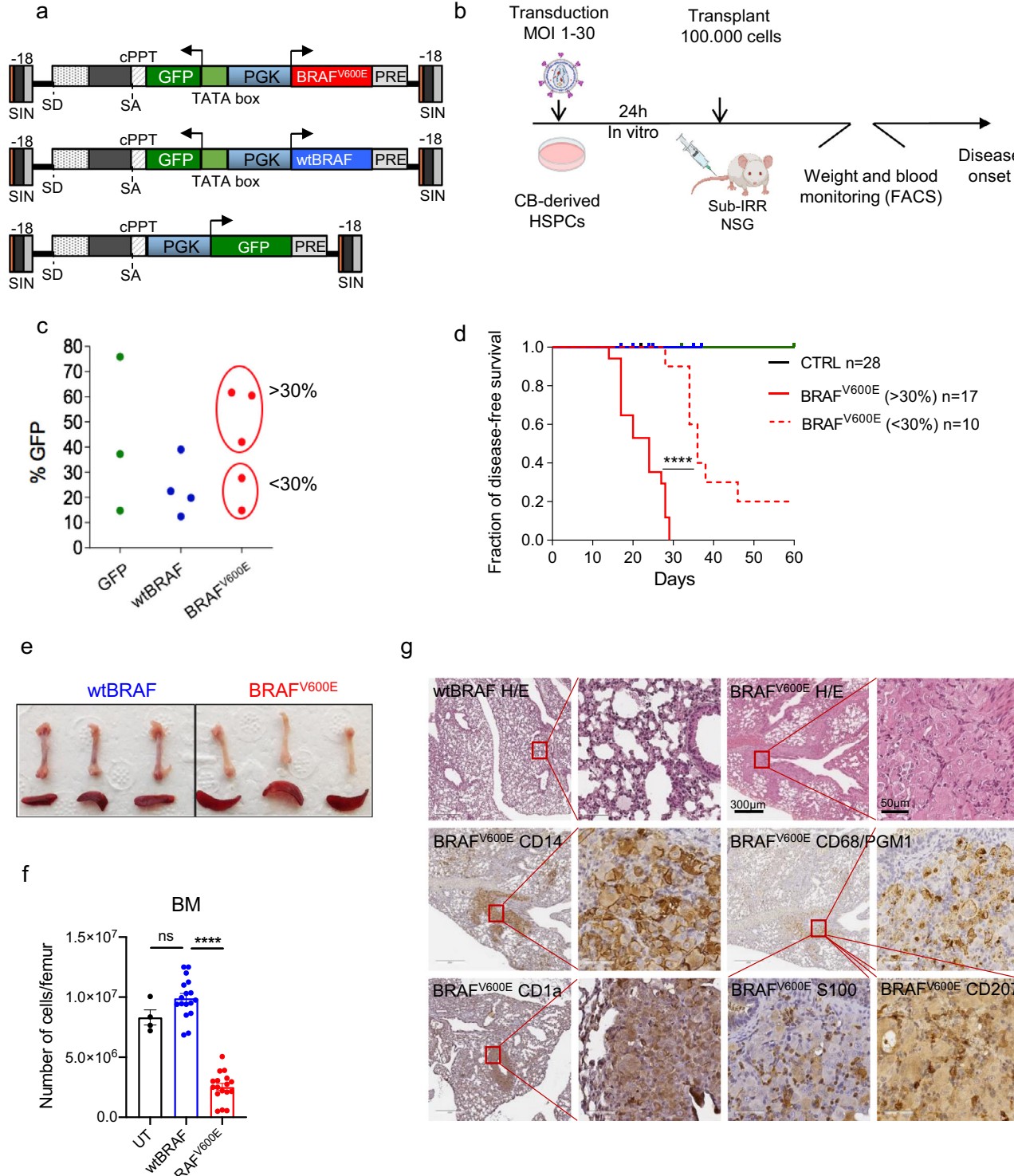

**Fig. 1 BRAF$^{V600E}$ activation in human HSPCs leads to lethal bone marrow aplasia and histiocytosis.** Schematic representation of lentiviral vectors (**a**) and experimental strategy (**b**) employed. **c** In vitro transduction level of GFP (green), wtBRAF (blue), and BRAF$^{V600E}$-expressing HSPCs (red). **d** Disease-free survival curves from CTRL group (mice receiving untransduced HSPCs (black, $n = 3$)), GFP only (green, $n = 12$), wtBRAF-expressing HSPCs (blue, $n = 13$), and mice transplanted with BRAF$^{V600E}$-expressing HSPCs (red) at different transduction levels (<30%, dashed line, $n = 10$ and >30%, continuous line, $n = 17$). Statistical test: log-rank (Mantel−Cox) test between survival curves of the low (<30% BRAF$^{V600E}$-expressing HSPCs) and high (>30% BRAF$^{V600E}$-expressing HSPCs) dose BRAF$^{V600E}$ groups. ****$p < 0.0001$. **e** Representative images of femur and spleen from mice transplanted with *wtBRAF* and BRAF$^{V600E}$-expressing HSPCs at the time of sacrifice. **f** Number of cells in femurs of mice transplanted with untransduced (UT, black, $n = 4$), wtBRAF (blue, $n = 17$) or BRAF$^{V600E}$ (red, $n = 17$)-expressing HSPCs. Data presented as mean values $+/−$ SEM. Statistical test: Kruskal−Wallis with Dunn's multiple comparisons ****$p = 0.0001$; ns non significant. **g** Immunohistochemical analysis of a lung lesion from a representative mouse transplanted with BRAF$^{V600E}$-expressing HSPCs at the time of sacrifice. Lung morphology analyzed by hematoxylin and eosin staining. A lung of a mouse transplanted with wtBRAF-expressing HSPCs was used as control. Markers for immunohistochemistry include: CD14 for monocytes, CD1a for myeloid dendritic cells, CD68/PGM1 for large macrophages, S100 for dendritic cells, and CD207 for smaller dendritic cells. Scale bars: 50 or 300 μm. ($n = 4$ mice for wtBRAF and $n = 3$ mice for BRAF$^{V600E}$ group).

frequently BRAF$^{V600E}$, have been reported in patients with hairy-cell leukemia (HCL), a B cell malignancy[6], and in a heterogeneous group of histiocytoses, inflammatory clonal neoplasms of dendritic cells, and macrophages[7]. The heterogeneity and aggressiveness of hematologic diseases consequent to MAPK pathway activation have been linked to the occurrence of BRAF$^{V600E}$ mutations in hematopoietic cell types at distinct differentiation stages[8]. In particular, BRAF$^{V600E}$ activation in bone marrow-derived HSPCs[9] correlates with the most aggressive forms of histiocytosis, characterized by dismal patient outcome and compromised functionality in the BM, liver, and spleen[10,11]. Several genetically engineered mouse models of Braf$^{V600E}$ expression in pan or restricted hematopoietic cell lineages have shown a spectrum of diverse phenotypes ranging from histiocytosis-like disease to HCL[12,13]. Unfortunately, these mouse models do not fully recapitulate the heterogeneity observed in human histiocytic lesions and did not provide mechanistic insights on the role of OIS on disease etiology and progression. Furthermore, there are fundamental differences in how the senescence program is controlled and executed in mouse versus human cells and it was reported that the response to senescence triggers may differ between the two species[14].

Here, we developed a humanized mouse model in which BRAF$^{V600E}$-expressing human HSPCs are transplanted into *NOD.Cg-Prkdc$^{scid}$Il2rg$^{tm1Wjl}$* (NSG) immunocompromised mice and investigated the impact of OIS in the human hematopoietic system. Forced BRAF$^{V600E}$ expression in human HSPCs resulted in a severe skewing towards the monocytic lineage due to a marked impairment in lymphoid output. Mice developed aggressive histiocytosis with multi-systemic infiltration of senescent macrophages and dendritic cells characterized by an activated inflammatory program sustained by senescence-associated cytokines. Lesions were not only composed by senescent BRAF$^{V600E}$-expressing cells but also by non-mutated bystander histiocytes that acquired paracrine senescence features mainly via TNFα production. Our findings establish a paradigm of pathogenesis of high-risk inflammatory neoplasms in which disease onset is preceded by the establishment of OIS in human hematopoiesis and point to the innovative therapeutic potential of senescence modulation for disease treatment.

## Results

**BRAF$^{V600E}$ activation in human HSPCs leads to lethal bone marrow aplasia and histiocytosis.** To study the impact of oncogenic BRAF activation on human HSPCs, human/mouse hematochimeras were generated by transplanting immunocompromised (NSG) mice with human cord blood (CB)-derived CD34$^+$ cells engineered with a lentiviral vector (LV)-expressing coordinately BRAF$^{V600E}$ and GFP (hereafter named BRAF$^{V600E}$)

under the control of a bidirectional human phosphoglycerate kinase (*hPGK*) promoter. As controls, we employed cells transduced with a bidirectional LV-expressing the wild-type form of human BRAF together with GFP (hereafter named wtBRAF), a LV expressing only GFP (hereafter named GFP) or untransduced cells (UT) (Fig. 1a). After transplantation mice were monitored for hematopoietic reconstitution and health status (Fig. 1b). We transduced HSPCs at different vector doses (multiplicity of infection (MOI) = 1−30) in order to reach transduction levels ranging from 10 to 60% of transgene-expressing cells, thus covering different proportions of BRAF$^{V600E}$-expressing hematopoietic progenitors (Fig. 1c).

Upon transplantation, only animals receiving BRAF$^{V600E}$-expressing HSPCs (BRAF$^{V600E}$ group) showed evident body weight loss, coupled with other signs of illness such as kyphosis, reluctance to move, and tremors. The median time of disease onset was dependent on the proportion of BRAF$^{V600E}$-expressing cells since mice transplanted with <30% of transgene-expressing cells showed disease symptoms significantly later compared to mice transplanted with >30% of BRAF$^{V600E}$-expressing cells (36 days vs 25.5 days median disease-free survival from transplant, $p < 0.001$) (Fig. 1d). Conversely, none of the mice transplanted with UT or wtBRAF and GFP-expressing HSPCs developed signs of malaise. To perform comparative analyses at matched time points, when mice transplanted with BRAF$^{V600E}$-expressing cells developed signs of disease, a similar number of control mice were euthanized.

Macroscopic analysis of the internal organs at euthanasia showed markedly paler long bones in mice from the BRAF$^{V600E}$ group, suggesting a profound alteration of BM cellularity, accompanied by a modest spleen enlargement (Fig. 1e). No macroscopic alterations were observed in any other organ. The overall BM cellularity of mice transplanted with BRAF$^{V600E}$ transduced HSPCs was significantly reduced when compared to mice transplanted with untransduced or wtBRA F-expressing HSPCs (Fig. 1f). This reduction in BM cellularity in the mice of the BRAF$^{V600E}$ group involved both murine and human CD45$^+$ cells (Supplementary Fig. 1a, b) and was associated to a modest, although significant, increase of necrotic cells compared to controls (Supplementary Fig. 1c). Histological analysis of mice transplanted with BRAF$^{V600E}$-expressing cells revealed massive BM aplasia with multifocal infiltration of foamy human histiocytes (from grade 1 to 3) in various organs, including lung, liver, spleen, kidney, and central nervous system. Infiltrating histiocytes varied in morphology and human surface-marker profiles and could be distinguished into large macrophages (CD14$^+$, CD68$^+$), CD1a$^+$/S100$^+$ dendritic cells, and smaller CD207$^+$ dendritic cells (Fig. 1g and Supplementary Fig. 1d). Thus, the phenotype of myeloid cells was not univocal and different subpopulations of aberrant macrophages and dendritic

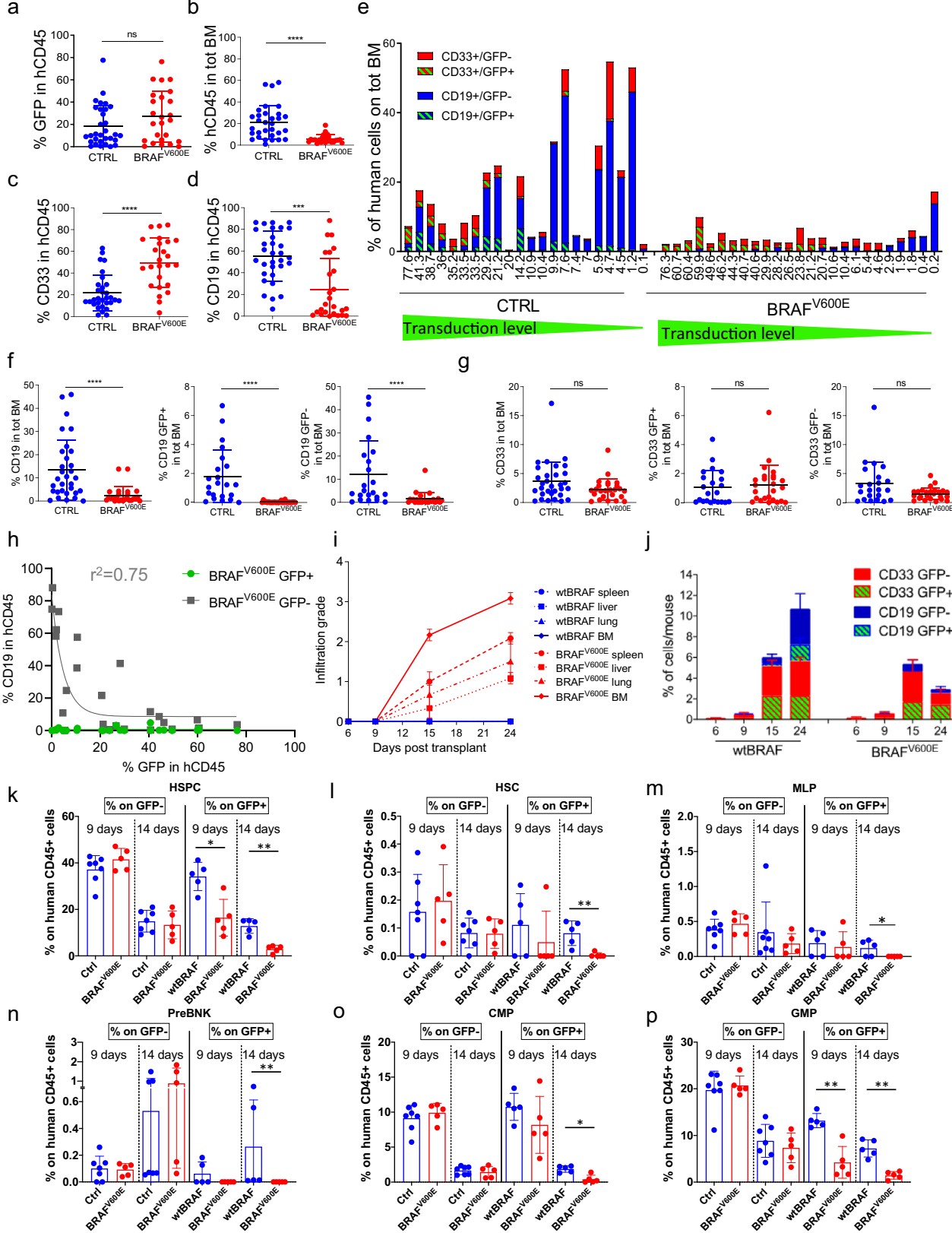

cells coexisted inside the lesions, a phenotype that is reminiscent of aggressive forms of human mixed histiocytosis.

Overall, our data indicate that transplantation of BRAF$^{V600E}$-expressing human HSPCs induces a dose-dependent lethal phenotype reminiscent of aggressive histiocytoses.

**BRAF$^{V600E}$ expression in human HSPCs results in a dose-dependent lymphoid impairment**. We next evaluated the impact of BRAF$^{V600E}$ expression on human hematopoietic engraftment and blood lineage reconstitution. Despite comparable levels of transduction efficiencies among experimental groups (as assessed by

**Fig. 2 BRAF$^{V600E}$ expression in human HSPCs results in a dose-dependent lymphoid impairment.** Percentage of (**a**) GFP$^+$ cells within the human graft (hCD45$^+$), (**b**) human engrafted cells in total BM and (**c**) myeloid (CD33$^+$) and (**d**) lymphoid B cells (CD19$^+$) within the human graft at euthanasia, in the BM of mice transplanted with CTRL (wtBRAF or GFP only, blue, $n = 31$) and BRAF$^{V600E}$ (red, $n = 25$) cells. Data are presented as mean +/− SD. Statistical test: Two-tailed Mann−Whitney. **e** Percentage of myeloid GFP$^+$ (red, dashed green) and myeloid GFP$^-$ (red), B (CD19$^+$) cells GFP$^-$ (blue) and B cells GFP$^+$ (blue, dashed green) within total BM from mice transplanted with CTRL or BRAF$^{V600E}$ HSPCs. Values on $x$-axis were ordered by transduction level. Percentage of GFP$^+$ and GFP$^-$ (**f**) B cells or (**g**) myeloid cells within total BM from mice transplanted with CTRL (blue, $n = 31$ for left panel, $n = 22$ for mid and right panel) and BRAF$^{V600E}$-expressing (red, $n = 25$) HSPCs. Data are represented as mean values +/− SD. Statistical test: two-tailed Mann−Whitney. **h** Percentage of GFP$^+$ and GFP$^-$ CD19$^+$ cells with respect to CD45$^+$/GFP$^+$ in each mouse. Exponential regression analysis. The goodness of fit measured as $R^2$. **i** Time-course of infiltration grade in organs from mice of the wtBRAF (blue) and BRAF$^{V600E}$ (red) groups; $n = 3$ for each time point. Data are presented as mean +/− SD. Spleen: dashed line, circles; liver: dashed line, squares; lung: dashed line, triangles; BM: continuous line, dots. **j** Time-course analyses of the percentage of GFP$^+$ and GFP$^-$ myeloid and B cells in BM of mice transplanted with wtBRAF or BRAF$^{V600E}$ HSPCs; $n = 3$ each time point. Data are presented as mean +/− SD, see (**e**) for color code. **k**–**p** Percentage of human HSPCs subpopulations within GFP$^-$ and GFP$^+$ cells in BM cells harvested from mice at 9 and 14 days after transplant with CTRL (untransduced, blue, $n = 2$ for each time point), wtBRAF (blue, $n = 5$ for each time point) and BRAF$^{V600E}$ (red, $n = 5$ for each time point) HSPCs. Data are presented as mean +/− SD (HSPC hematopoietic stem and progenitor cells; HSC hematopoietic stem cells; MLP multi-lymphoid progenitors; PreBNK B and NK cell progenitors; CMP common-myeloid progenitors; GMP granulocyte/ monocyte progenitors). Statistical analysis between groups by two-tailed Mann−Whitney test. ns non-significant; *$p < 0.05$; **$p < 0.01$; ***$p < 0.001$; ****$p < 0.0001$.

GFP expression) (Fig. 2a), mice from the BRAF$^{V600E}$ group showed a significantly lower percentage of human hematopoietic cells (hCD45$^+$) in the BM compared to controls (Fig. 2b and Supplementary Fig. 2a). Within the human graft, we observed skewed hematopoiesis with a significant increase in the relative percentage of myeloid cells (CD33$^+$), accompanied by a significant reduction of B cell output (CD19$^+$) across mice (Fig. 2c, d and Supplementary Fig. 2b). Of note, B cells are the major human HSPC output in this NSG model, which helps understanding the dramatic effect on total cell number despite limited changes in myeloid output. When focusing on the impact of oncogene-expressing human cells on murine hematopoiesis, the relative percentage of the murine myeloid output in the mice of the BRAF$^{V600E}$ group compared to the other groups showed a marked skewing towards the monocytic/ macrophagic CD11$^{+high}$/Gr1$^-$/mCD45$^+$ lineage but did not involve granulocytes CD11$^{+high}$/Gr1$^+$/mCD45$^+$ (Supplementary Fig. 2c, d). We next analyzed the proportion of engrafted human myeloid and lymphoid outputs with respect to the total BM, thus taking into account murine and human cells. Across individual mice of the BRAF$^{V600E}$ group we observed a much lower percentage, and an overall number of CD19$^+$ B cells, which involved not only BRAF$^{V600E}$-expressing cells but also CD19$^+$ B cells that did not express the transgene (Fig. 2e). This lymphoid impairment of untransduced non-oncogene-expressing B cells was dependent on the overall number of BRAF$^{V600E}$-expressing cells, since mice transplanted with only a small fraction of BRAF$^{V600E}$-expressing HSPCs still had measurable levels of B cells (Fig. 2e). Statistical analysis confirmed the observed marked B cell impairment in both GFP$^+$ and GFP$^-$ fraction of the BRAF$^{V600E}$ group, while the percentage of myeloid (CD33$^+$) cells did not change compared to controls (Fig. 2f, g). To better investigate the impact of BRAF$^{V600E}$ overexpression on the B-cell lineage we correlated the percentage of oncogene and non-oncogene-expressing CD19$^+$ cells (GFP$^+$ and GFP$^-$, respectively) with the percentage of transduced human CD45$^+$ cells (hCD45$^+$GFP$^+$) in mice from the BRAF$^{V600E}$ group at the time of euthanasia. We found that CD19$^+$GFP$^+$ cells were absent from any analyzed mouse, even when the transduced cells within the human graft were rare (Fig. 2h). On the other hand, the GFP$^-$ fraction of CD19$^+$ B cells showed a strong negative correlation with the percent of overall hCD45$^+$GFP$^+$ cells. This non-cell-autonomous impairment was non-linear and resulted in a pronounced reduction of GFP$^-$ B cells even at transduction levels as low as 5−10%, best fitting with a one-phase decay function ($r^2 = 0.75$) (Fig. 2h). These results indicate that BRAF$^{V600E}$ expression in HSPCs is highly detrimental to lymphoid reconstitution in both cell-autonomous and non-cell-autonomous ways.

To study the dynamics of histiocyte dissemination we transplanted a second cohort of mice with a 30% fraction of transduced wtBRAF or BRAF$^{V600E}$-expressing HSPCs and progressively euthanized them at distinct time points. Starting from 9 days post-transplant sporadic abnormal histiocytes were detected in organs of mice from the BRAF$^{V600E}$ group. At 15 days after transplant, mice from the BRAF$^{V600E}$ group showed detectable grade 1−2 multi-organ histiocytic lesions that progressed up to grade 3 at day 24. Among different organs, BM showed the highest infiltration grade at any time point analyzed. Instead, no abnormalities were observed in mice from the wtBRAF group (Fig. 2i). We then studied the kinetics of hematopoietic reconstitution and performed immune-phenotypic characterization of CD33$^+$, CD19$^+$, and CD34$^+$ cells at 6, 9, 15, and 24 days after transplant (Fig. 2j and Supplementary Fig. 2e−h). This time-course analysis showed in the BM of mice of both control and BRAF$^{V600E}$ groups a steady increase of hCD45$^+$ cells, with comparable myeloid (CD33$^+$) and lymphoid (CD19$^+$) composition up to 15 days after transplant (Fig. 2j and Supplementary Fig. 2e, f). However, 24 days after transplant, mice from the BRAF$^{V600E}$ group showed a reduction in the percentage of hCD19$^+$ B cells in BM compared to controls (Fig. 2j), which was mainly attributed to the strong cell and non-cell-autonomous B cell impairment. Conversely, the percentage of CD33$^+$ myeloid cells was only slightly reduced (Fig. 2j and Supplementary Fig. 2e, f). Moreover, at 24 days post-transplant we observed a reduction in the fraction of oncogene-expressing myeloid progenitors (CD34$^+$/CD33$^+$) as well as in CD34$^+$ cells compared to controls. On the other hand, the GFP$^-$ fraction of CD34$^+$/CD33$^+$ and CD34$^+$ cells showed an expansion at 15- and 24-days post-transplant in both the wtBRAF and the BRAF$^{V600E}$ experimental groups, although with a higher extent in the BRAF$^{V600E}$ group (Supplementary Fig. 2g, h). These results indicate a different response of distinct hematopoietic cell types to oncogene activation in a cell-autonomous and a non-cell-autonomous fashion.

To delve deeper into the impact of BRAF$^{V600E}$ expression on human hematopoiesis, we applied the Whole Blood Dissection (WBD) technology[15] to BM-derived cells from BRAF$^{V600E}$ or wtBRAF transplanted mice at 9 and 14 days after transplant, before the onset of hematological abnormalities (Supplementary Fig. 3a). The transduction level in this new set of transplantation experiments was around 10% for the BRAF$^{V600E}$ experimental group and 25% for the wtBRAF group (Supplementary Fig. 3b). Vector marking levels of the different progenitor populations varied depending on the permissiveness to transduction of the specific progenitor lineages. The most transduced (>80%) HSPC subsets were the megaKaryocyte progenitors (MKP) and

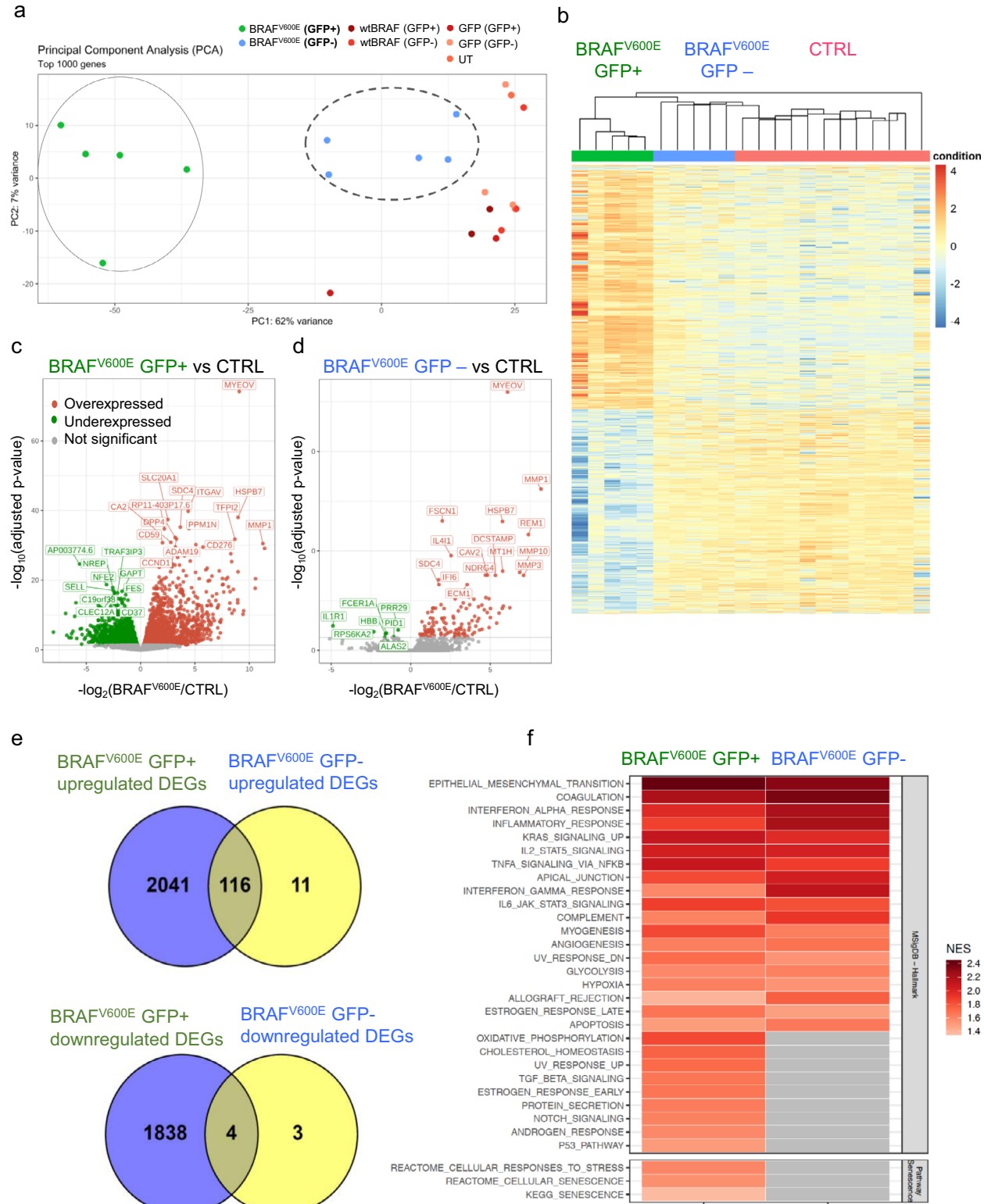

megakaryocyte/erythrocyte progenitors (MEP), followed by the HSC-enriched fraction (CD90$^+$), erythroid progenitors (EP), common-myeloid progenitors (CMP), and multi-potent progenitors (MPP) with marking levels ranging from 15 to 30%, while multi-lymphoid progenitors (MLP) and B and NK cell progenitors (PreBNK) had less than 5% marking (Supplementary Fig. 3c). Of note, transduction levels across hematopoietic subsets did not vary significantly between the wtBRAF and BRAF$^{V600E}$

groups. We found that mice transplanted with BRAF$^{V600E}$-expressing cells showed a reduced HSPC content within the GFP$^+$ fraction already at 9 days post-transplant (Fig. 2k). Moreover, BRAF$^{V600E}$-expressing cells were progressively impaired among all HSPC subpopulations, including the most primitive HSCs, lymphoid progenitors (MLP and PreBNK), and myeloid progenitors (CMP and GMP) as well as in the erythroid/megakaryocyte compartment (MKP, MEP, EP, and ETP) with more pronounced

**Fig. 3 Global gene expression analysis upon oncogene-activation. a** Principal component analysis (PCA) of transcriptional profiles of sorted human myeloid cells (CD33$^+$) GFP$^+$ and GFP$^-$ fractions from control mice, transplanted with human HSPCs transduced with GFP (light pink for GFP$^-$ and dark pink for GFP$^+$), wtBRAF (light red for GFP$^-$ and dark red for GFP$^+$) or untransduced cells (salmon), and mice transplanted with BRAF$^{V600E}$-expressing HSPCs at 16 days post-transplantation (green for GFP$^+$ and blu for GFP$^-$ cells). **b** Unsupervised clustering analysis of DEGs in the different experimental groups. **c, d** Volcano plots highlighting significant downregulated (green) and upregulated (red) genes in sorted human myeloid cells (CD33$^+$) from the BRAF$^{V600E}$ mice group (both GFP$^{+/-}$ fractions) compared to controls. The gene symbols of the top 15 upregulated and the top ten downregulated genes based on adjusted $p$-value are indicated. Statistical test: two-sided exact test for negative-binomial distribution, the Benjamini and Hochberg's approach has been used for controlling the false discovery rate (FDR). **e** Venn Diagram representation of upregulated and downregulated DEGs in GFP$^+$ and GFP$^-$ CD33$^+$ cells of the BRAF$^{V600E}$ group showing that about 90% of the DEGs in the GFP$^-$ fraction were also deregulated in the GFP$^+$ fraction. **f** Gene set enrichment analysis (GSEA) of sorted CD33$^+$/GFP$^+$ and CD33$^+$/GFP$^-$ cells from the BRAF$^{V600E}$ group. Normalized enrichment score (NES) is indicated.

effects at 14 days post transplantation, thus anticipating the impairment observed in the more differentiated counterparts (Fig. 2l−p and Supplementary Fig. 3d−g). The only exception to the general HSPC impairment was represented by the fraction of MPP, which significantly increased at 9 days post-transplant in the BRAF$^{V600E}$ group (Supplementary Fig. 3h; see also Supplementary Fig. 4 for the gating strategy used to identify all HSPC subsets). Instead, there was no difference in the frequency of total HSPCs and in the distribution of HSPC subpopulations within the GFP$^-$ fraction from both wtBRAF and BRAF$^{V600E}$ groups. The lack of the non-cell-autonomous effect of BRAF$^{V600E}$ expression in this experimental setting was expected given the lower levels of HSPC transduction compared to previous experiments and the shorter time point of analysis.

Altogether, our findings indicate that BRAF$^{V600E}$ expression causes a profound impairment in human primitive HSPCs leading to progenitor and B-cell reduction, which over time, affects also untransduced cells, indicating both cell-intrinsic and non-cell-autonomous detrimental effects of BRAF$^{V600E}$ expression on human hematopoiesis.

**Global gene expression analysis unravels a cell- and non-cell-autonomous inflammatory program upon oncogene activation in HSPCs.** To better study the cell- and non-cell-autonomous effects of BRAF$^{V600E}$ expression on human hematopoiesis, we performed whole transcriptomic analysis on FACS-purified human myeloid hCD45$^+$/CD33$^+$/GFP$^+$ and hCD45$^+$/CD33$^+$/GFP$^-$ cells collected at 16 days post transplantation from mice of the BRAF$^{V600E}$ and control groups (Supplementary Fig. 5a). Principal component and hierarchical clustering analysis of gene expression datasets revealed that the overall transcriptome of myeloid cells from UT, wtBRAF, or GFP groups (CTRL) was similar and clustered together, irrespectively of their vector marking status and therefore was used as a unified control group in subsequent analyses (Fig. 3a, b). Conversely, the transcriptome of both transduced and untransduced cells from mice transplanted with BRAF$^{V600E}$-expressing HSPCs clustered independently from controls (Fig. 3a, b). BRAF$^{V600E}$-expressing cells (GFP$^+$) displayed 3999 significantly differentially expressed genes (DEGs) (FDR < 0.05) (2157 upregulated and 1842 downregulated), and GFP$^-$ cells from mice of the BRAF$^{V600E}$ group showed 134 DEGs (127 genes upregulated and 7 genes downregulated) compared to controls (Fig. 3c, d). Of note, out of 134 DEGs found in the GFP$^-$ cells from the BRAF$^{V600E}$ group, 120 genes (around 90%) were coherently regulated in BRAF$^{V600E}$-expressing cells, indicating the activation of a shared transcriptional program between BRAF$^{V600E}$-expressing and bystander cells (Fig. 3e and Supplementary Data File).

Gene set enrichment analysis (GSEA) against hallmark gene sets from Molecular Signatures Database (v6.2) and the selected senescence-related gene lists from REACTOME and KEGG databases, showed that the DEGs identified in GFP$^+$ cells from the BRAF$^{V600E}$ group were significantly enriched for categories involved in inflammatory responses (such as TNFα signaling via NFkB, IL2-STAT5 signaling, IL6-JAK-STAT3 signaling, IFNγ

and IFNγ responses), metabolic activation (oxidative phosphorylation, glycolysis, cholesterol homeostasis), protein synthesis and secretion, DNA damage response (DDR) (p53 pathway, UV-response), mitogenic signaling (KRAS and NOTCH signaling), tissue invasion (epithelial to mesenchymal transition, apical junctions) and cellular senescence (Fig. 3f and Supplementary Fig. 5b, c). GSEA on DEGs identified in GFP$^-$ cells showed a significant enrichment in inflammatory response pathways (TNFα signaling via NFkB, IL2-STAT5 signaling, IL6-JAK-STAT3 signaling, IFNγ response) similarly to GFP$^+$ oncogene-expressing cells (Fig. 3f).

Altogether, these results support the establishment of a cell-autonomous cellular program upon BRAF$^{V600E}$ expression in HSPCs, which is characterized by DDR induction, expression of cell-cycle inhibitors, and activation of a complex network of inflammatory cytokines, chemokines, and MMPs. These global transcriptional changes are in line with the establishment of an oncogene-induced senescence phenotype in BRAF$^{V600E}$-expressing myeloid cells that significantly alters the transcriptional program of bystander cells through activation of an inflammatory gene signature.

**BRAF$^{V600E}$ expression in HSPCs promotes oncogene-induced senescence.** Given the results of our transcriptomic analyses, we further investigated the establishment of a senescence program either in vivo or in vitro upon oncogene expression in HSPCs. In BM cells harvested at euthanasia from mice of the BRAF$^{V600E}$ group, we detected an increase in the frequency of cells expressing the senescence-associated cell cycle inhibitor p16$^{INK4A}$ (hereafter named p16; Fig. 4a, b). The increased frequency of p16$^+$ cells was also observed upon BRAF$^{V600E}$ expression in in vitro transduced HSPCs (with 30% of vector marked cells) after 4 days post-transduction (Fig. 4c, d). Human CD45$^+$ cells harvested from the BM of transplanted mice, as well as in vitro cultured HSPCs, when transduced with the BRAF$^{V600E}$-expressing LV, displayed a higher frequency of senescence-associated-β-galactosidase (SA-β-gal) positive cells compared to controls (Supplementary Fig. 6a–c). Consistently with the establishment of a senescence program, we also detected a higher percentage of cells displaying 53BP1 nuclear foci in BRAF$^{V600E}$-expressing HSPCs (Fig. 4e, f) and increased levels of hCD45$^+$ cells positive for the phosphorylated form of the DNA Damage Response protein Ataxia Telangiectasia Mutated (pATM) (Supplementary Fig. 6d). We also observed the upregulation at the mRNA level of *p16* and *p21$^{CIP1}$* (hereafter named p21; Supplementary Fig. 6e, f) and SASP factors such as *MCP1*, *IL8*, and *IL1α* and *IL1β* (Supplementary Fig. 6g−j). As previously reported for oncogene expression in human fibroblasts[3], we also found that within the first 48 h of liquid culture BRAF$^{V600E}$-expressing HSPCs showed a significantly higher proliferation rate compared to wtBRAF transduced cells (Fig. 4g), as also indicated by the statistical analysis performed with a linear mixed-effects model (LME) and slowed down their proliferation at 120 h post transduction, reaching a marked proliferation delay at 216 h in accordance with the establishment of a

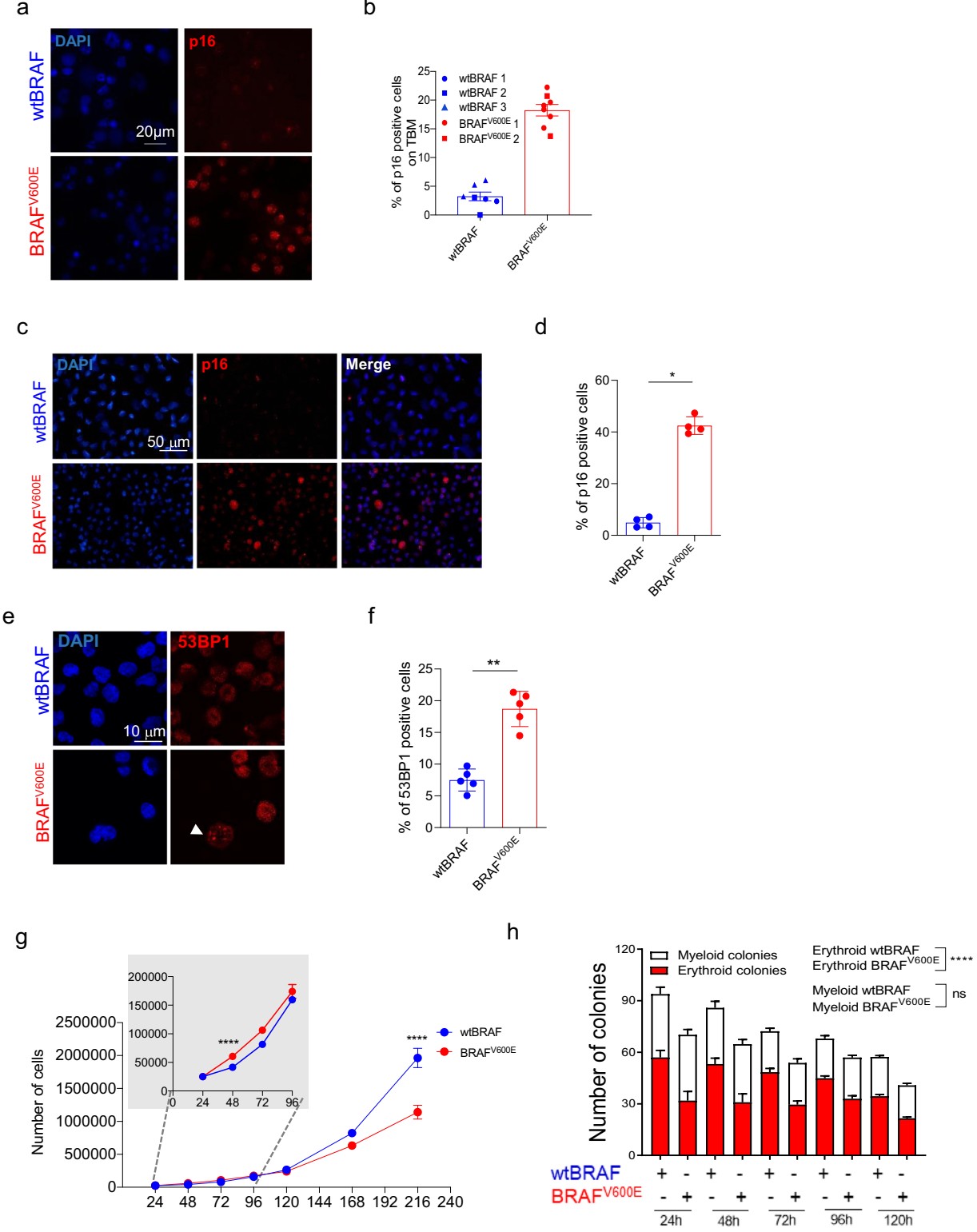

senescence program in the absence of overt apoptosis (Fig. 4g and Supplementary Fig. 6k). Moreover, BRAF^V600E-expressing HSPCs, when plated on methylcellulose at different time points of in vitro culture, showed a significant reduction in colony-forming potential overtime compared to controls as shown by LME model (Fig. 4h). The impairment in colony-forming potential was mainly

due to a significant reduction in erythroid (red) colonies while the number of myeloid (white) colonies remained unaltered compared to wtBRAF controls (Fig. 4h). Activation of cell cycle inhibitors was associated to dramatic changes in cell morphology of oncogene-expressing (GFP^+) HSPCs, as well as in untransduced (GFP^-) cells from the BRAF^V600E group (Supplementary Fig. 7a).

**Fig. 4 BRAF$^{V600E}$ expression in HSPCs induces p16 expression and cell cycle arrest. a** Representative images of p16 (red) and DAPI (blue) positive cells in BM of mice transplanted with wtBRAF and BRAF$^{V600E}$-expressing HSPCs at the time of euthanasia. Scale bar: 20 μm. **b** Quantification of p16$^+$ cells from (A), wtBRAF (blue, $n = 3$), and BRAF$^{V600E}$ (red, $n = 2$) groups. Each symbol indicates independent measurements from a given mouse. Lines indicate mean values ± SEM. **c** Representative images of p16 (red) and DAPI (blue) positive in wtBRAF and BRAF$^{V600E}$-expressing HSPCs 120 h after in vitro transduction. Scale bar: 50 μm. **d** Quantification of p16$^+$ cells from (**c**); wtBRAF (blue) and BRAF$^{V600E}$ (red): $n = 4$. Lines indicate mean values ± SEM. Statistical test: two-tailed Mann−Whitney. *$p < 0.05$. **e** Representative images of 53BP1 foci (red) and DAPI (blue) in wtBRAF and BRAF$^{V600E}$-expressing HSPCs 120 h post-transduction. **f** Quantification of 53BP1 foci from (**e**); wtBRAF (blue, $n = 5$); BRAF$^{V600E}$ (red, $n = 5$). Lines indicate mean values ± SEM. Statistical test: Two-tailed Mann−Whitney **$p < 0.01$ ($p = 0.0079$). **g** Growth curves of wtBRAF or BRAF$^{V600E}$-expressing HSPCs. Cells were counted at the indicated time points after transduction. wtBRAF ($n = 8$ repeated measurements for four independent donors); BRAF$^{V600E}$ ($n = 8$ repeated measurements for four independent donors). Data are presented as mean +/− SD. Linear mixed-effects model (LME) followed by post hoc analysis at 48 and 216 h (****$p < 0.0001$). Inset indicates growth curves of BRAF$^{V600E}$ compared to wtBRAF at early time points. See also Source Data file. **h** Clonogenic potential of wtBRAF and BRAF$^{V600E}$-expressing HSPCs in methylcellulose assays over time. wtBRAF ($n = 8$ repeated measurements for four independent donors); BRAF$^{V600E}$ ($n = 8$ repeated measurements for four independent donors). Data are presented as mean values +/− SEM. Linear mixed-effects model (LME). ****$p < 0.0001$. See also Source Data file. All graphs in this figure display data from wtBRAF in blue and from BRAF$^{V600E}$ in red.

To better characterize senescence establishment in the histiocytic infiltrates, we performed immunohistochemical analysis of serial sections of lung lesions in mice transplanted with BRAF$^{V600E}$-expressing HSPCs, detecting vector marked GFP$^+$ cells, p16$^+$ cells, and the myeloid lineage markers S100, CD207, and PGM1. From this analysis, we found that GFP and p16 were expressed together at high frequency and co-expressed different myeloid markers. However, not all histiocytes within the lesions were vector marked nor p16 positive, and conversely, several GFP$^−$ cells were p16 positive. The pathological analysis of the infiltrates confirmed that the morphological phenotype of the histiocytes within the lesions was heterogeneous and resembled those observed in human mixed histiocytosis (Fig. 5a). Immunohistochemical analysis on lung sections harvested at different time points after transplantation showed sporadic p16$^+$ cells as early as 9 days post transplantation in one out of three mice from the BRAF$^{V600E}$ group (Fig. 5b). The number of p16$^+$ cells progressively increased at 24 days after transplant in all mice transplanted with BRAF$^{V600E}$-expressing HSPCs (Fig. 5b). Most p16$^+$ cells localized into tissue lesions composed by enlarged human histiocytes and showed mutual exclusivity with the proliferation marker Ki67 (Fig. 5b). FACS analysis showed that myeloid cells (CD33+) from the BM of mice transplanted with BRAF$^{V600E}$-expressing HSPCs were significantly enriched for p16$^+$ and SA-βGal$^+$ cells compared to controls. This increase in senescent cells was also observed in the GFP$^−$ fraction of the human graft, further supporting the establishment of senescence in bystander cells (Fig. 5c, d and Supplementary Fig. 7b).

The accumulation of cellular senescence markers such as p16, SA-β-gal, and the lack of the proliferation marker Ki67 was detected also in skin and meninges histiocytic lesions from patients affected by BRAF$^{V600E}$-mutated Erdheim-Chester histiocytosis (Fig. 5e and Supplementary Fig. 7c). Interestingly, while the number of mutated BRAF$^{V600E}$-cells (as detected by VE1 staining) was limited, the p16 signal was widespread and involved almost all the cells within the patient lesion, thus suggesting that both oncogene and non-oncogene-expressing cells accumulate senescence markers also in patient samples (Fig. 5e).

To dissect the contribution of p16-mediated cell cycle arrest to the observed phenotype of senescence induction in BRAF$^{V600E}$-expressing HSPCs, we first downregulated the levels of p16/p19$^{ARF}$ by RNA interference (hereafter defined as short hairpin (sh) RNA p16), then transduced HSPCs with wtBRAF or BRAF$^{V600E}$-expressing vectors and evaluated their clonogenic output, proliferation and senescence emergence (Fig. 6a). In this experimental setting, we observed that p16 inhibition resulted in the reduction of both cell cycle inhibitors *p16* and *p21* (Fig. 6b, c), rescued the proliferation block of BRAF$^{V600E}$-expressing HSPCs (Fig. 6d, e), and partially rescued their colony-forming capability (Fig. 6f) without affecting the production of SASP cytokines

(Fig. 6g−k). We then asked if we could dampen SASP cytokines by knocking down the master transcriptional regulator of pro-inflammatory cytokines NF-κB. To test this hypothesis, we first downregulated the levels of RELA/p65, an essential subunit of the NF-κB heterodimer, by RNA interference (hereafter defined as shp65), then transduced HSPCs with wtBRAF or BRAF$^{V600E}$-expressing vectors. When we evaluated clonogenic output, we observed a strong rescue of the clonogenic potential of oncogene-expressing HSPCs (Fig. 6l), reinforcing the idea that the HSPC clonogenic impairment observed in vitro upon BRAF$^{V600E}$ expression is aid by the production of SASP cytokines. In agreement with this, we also report a consistent trend of decrease in several SASP cytokines at mRNA level upon RELA/p65 downregulation, and a reduction in *p21* levels while no significant changes in *p16* expression levels were observed (Supplementary Fig. 7d). These data indicate that NFkB controls senescence establishment and SASP in oncogene-expressing HSPCs and that SASP suppression ameliorates HSPC clonogenicity.

Altogether, these results support the establishment of a p16 and NFkB dependent senescence program upon BRAF$^{V600E}$ expression in HSPCs, characterized by increased cell-cycle inhibitors, accumulation of SA-β-Gal activity, and clonogenic impairment. We also provide experimental evidence that senescence occurs in myeloid lesions from patients with histiocytoses and can be detected in both mutated and non-mutated cells.

**SASP factors from senescent HSPCs induce paracrine senescence in non-oncogene-expressing cells.** Since senescence is characterized by the activation of a pronounced pro-inflammatory secretory phenotype (SASP), we measured the protein levels of several SASP factors upon BRAF$^{V600E}$ expression in the blood plasma of transplanted mice and in vitro cultured HSPCs. From this analysis, we observed a significant increase of IL1α, IL1β, TNFα, IL8, IL6, MCP1, and CCL4 in BRAF$^{V600E}$ treatment group compared to controls, both in vivo and in vitro (Fig. 7a, b respectively) while the levels of IFNγ, IL12, and GM-CSF in the blood of BRAF$^{V600E}$ transplanted mice remained unaltered (Supplementary Fig. 8a). Importantly, the concentration of SASP-related human pro-inflammatory cytokines in the plasma of mice of the BRAF$^{V600E}$ group at two weeks after transplant inversely correlated with mice survival, representing a good predictor of disease aggressiveness (Fig. 7c). To investigate whether secreted pro-inflammatory cytokines produced by HSPCs undergoing OIS are responsible for the non-cell-autonomous transmission of senescence to bystander cells, we exposed untransduced HSPCs to conditioned media (CM) collected from in vitro cultured HSPCs expressing activated BRAF$^{V600E}$ or wtBRAF control and tested their proliferation, clonogenic potential and the appearance of senescence markers.

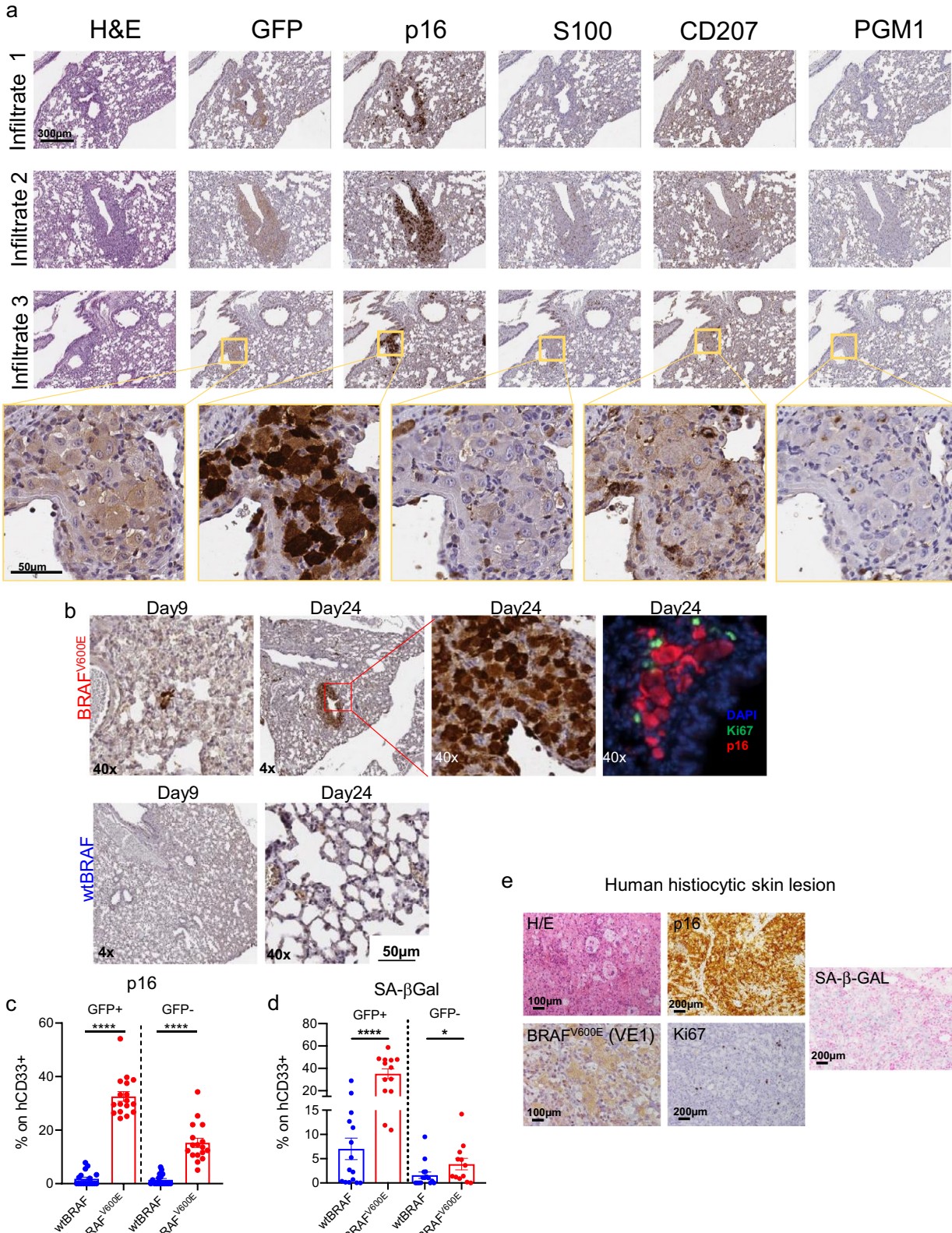

Cells exposed to CM from BRAF$^{V600E}$-expressing HSPCs cells showed a significant reduction in their proliferation rate overtime and clonogenic potential (Fig. 7d, e), accompanied by a significant increase in the gene expression levels of cell cycle inhibitors, *p16* and *p21*, and SASP cytokines, such as *IL1α*, *IL1β*, *TNFα*, *IL8*, and *MCP1*, when compared to HSPCs exposed to wtBRAF, derived CM (Supplementary Fig. 8b). To identify the main players involved in the paracrine transmission of senescence, we prioritized the testing of the inflammatory cytokines IL1α, IL1β, and TNFα, based on their role as regulators of the SASP inflammatory network[16,17], as well as on their relevance to the pathophysiology and targeted treatment of human inflammatory myeloid neoplasms. Thus, we exposed untransduced HSPCs to these recombinant human cytokines and tested their

**Fig. 5 BRAF$^{V600E}$-mediated oncogene-induced senescence in histiocytic lesions and in human grafts. a** Immunohistochemical analysis of serial sections of lungs lesions from a representative mouse transplanted with BRAF$^{V600E}$-expressing HSPCs at the time of sacrifice. Markers for immunohistochemistry include: GFP for vector marking, p16 for senescence/cell cycle arrest, S100, CD207, and PGM1 for dendritic cells and macrophages. Tissue morphology analyzed by hematoxylin and eosin (H&E). Yellow boxes indicate zoom-in areas. Scale bars: 300 or 50 µm. $n = 1$ representative mouse. **b** Representative immunohistochemical analysis for p16 expression in lung sections from mice transplanted with wtBRAF or BRAF$^{V600E}$-expressing HSPCs harvested at 9 and 24 days after transplantation. Immunofluorescence analysis for p16 (red) and the proliferation marker Ki67 (green) in lung lesions from a mouse of the BRAF$^{V600E}$ group at 24 days post-transplantation. ($n = 3$ for each time point). Nuclei stained by DAPI (blu). Scale bar: 50 µm. **c, d** Frequency (in percentage) of GFP$^+$ or GFP$^-$ human myeloid cells (CD33$^+$) which express the senescence markers p16 (**c**) or SA-β-gal (**d**). FACS analysis was performed on BM cells obtained at euthanasia from mice transplanted with wtBRAF (blue) or BRAF$^{V600E}$ (red) HSPCs ($n = 17$ each group). Data are presented as mean values +/− SEM. Statistical test: two-tailed Mann−Whitney ****$p < 0.0001$, *$p < 0.05$. **e** Representative images of a skin lesion from a patient with histiocytosis stained for p16 (positive ~85%), BRAF$^{V600E}$ (VE1) (positive ~25%), Ki67 (positive <1%), and SA-β-Gal (positive ~48%). Tissue morphology was analyzed by hematoxylin and eosin. Scale bars: 100 or 200 µm.

effects on proliferation, clonogenic potential, and senescence markers. We found that all three cytokines initially increased proliferation compared to controls (Supplementary Fig. 8c). Treatment with human recombinant IL1β at escalating doses did not affect HSPC proliferation in the long term (Fig. 7f), while IL1α treatment resulted in only a modest dose-dependent impairment of cell proliferation over-time (Fig. 7g, Supplementary Fig. 8c), as shown by LME model. Conversely, TNFα treatment induced a pronounced and significant dose-dependent cell cycle arrest (Fig. 7h) and was associated with changes in HSPC morphology (Supplementary Fig. 8d). Consistently with this effect, we observed a marked time and dose-dependent decline in clonogenic capacity of TNFα-treated HSPCs, with a greater reduction in the number of erythroid colonies compared to myeloid counterparts (Fig. 7i). We also detected a modest but significant reduction in colony-forming potential in HSPCs treated with the highest dose of IL1α but not upon IL1β treatment (Fig. 7i). Gene expression changes across different HSPC donors revealed that TNFα treatment induced the expression of the cell cycle inhibitor *p21* and downstream SASP effectors in a dose-dependent manner (Fig. 7j and Supplementary Fig. 8e). Of note, TNFα was also found highly expressed in the histiocytic lesion from the previously described BRAF$^{V600E}$-mutated Erdheim-Chester patient (Fig. 7k).

Altogether, these results support the establishment of a pro-inflammatory cytokine program upon OIS establishment in HSPCs, which negatively correlates with mice survival. Furthermore, we identified TNFα as an upstream mediator of senescence establishment and SASP production in HSPCs, pointing to its role in mediating senescence in bystander non-oncogene-expressing cells.

**TNFα inhibition rescues paracrine HSPC dysfunction upon oncogene activation.** To confirm the prevailing role of TNFα in the transmission of paracrine senescence, we inhibited TNFα activity in CM collected from BRAF$^{V600E}$-expressing HSPCs with infliximab, a chimeric monoclonal antibody that binds with high affinity to the soluble or transmembrane forms of TNFα, and addressed its impact on cell proliferation, clonogenic capacity and the appearance of senescence markers (Fig. 8a). TNFα blockade partially rescued the proliferative defects of HSPCs exposed to BRAF$^{V600E}$-derived CM (Fig. 8b) and reduced the expression of senescence-associated genes, such as cell cycle inhibitors *p16* and *p21* and SASP cytokines *IL1α, IL1β, TNFα, IL8,* and *MCP1,* to levels comparable to control groups (Fig. 8c and Supplementary Fig. 9a). Moreover, TNFα blockade ameliorated the erythroid output of HSPCs exposed to BRAF$^{V600E}$-CM in colony-forming assay (Fig. 8d). We then tested if TNFα inhibition in vivo could slow down disease progression in mice transplanted with BRAF$^{V600E}$-expressing HSPCs. Consistently with previous experiments, mice transplanted with BRAF$^{V600E}$-expressing HSPCs quickly developed disease symptoms accompanied by

severe weight loss and were euthanized between 16 and 22 days post-transplant (median survival = 17 days). Conversely, infliximab treatment significantly extended disease-free survival (median survival = 20.5 days) (Fig. 8e). FACS analysis on engrafting human CD45$^+$ cells in BM at euthanasia showed partial recovery of lymphoid lineage in mice from the BRAF$^{V600E}$ group treated with infliximab, with no effects on myeloid counterparts regardless of their marking status (Fig. 8f, g and Supplementary Fig. 9b, c). Importantly, infliximab-mediated lymphoid recovery in mice transplanted with BRAF$^{V600E}$-expressing HSPCs involved mainly non-oncogene expressing (GFP$^-$) cells and only modestly oncogene-expressing (GFP$^+$) cells (Fig. 8g).

In conclusion, our findings indicate that the blockade of the OIS-dependent SASP program by TNFα inhibition ameliorates disease symptoms, reduces paracrine senescence, and partially rescues the lymphoid impairment observed upon oncogene-activation in hematopoietic progenitors.

## Discussion

In this study we dissected the impact of the activated form of BRAF (BRAF$^{V600E}$) activation on HSPC biology and on the hematopoietic system by exploiting a humanized mouse model of transplantation of human HSPCs expressing BRAF$^{V600E}$, a variant frequently detected in histiocytoses, inflammatory hematological neoplasms. By integrating global transcriptomic analyses and cellular and immune-phenotypic characterization of lineage output we discovered that the activation of an oncogene-induced senescence program impairs HSPC function, features myeloid-restricted hematopoiesis, and leads to a widespread inflammatory condition that ultimately results in multi-organ toxicity and lethality. The senescence features identified in our model were also detected in myeloid lesions from a patient affected by histiocytosis, further highlighting the role of oncogene-induced senescence as an important trigger of the chronic inflammatory status with devastating consequences at the organismal level. Blockade of the senescence-associated inflammatory program by TNFα inhibition partially rescues the detrimental effects of senescence on human hematopoietic progenitors and ameliorates disease phenotype.

The first evidence for the detrimental effects of oncogene activation in HSPCs comes from the highly penetrant lethal phenotype observed in transplanted animals even in the presence of a limited fraction of BRAF$^{V600E}$-expressing cells. Lethality, however, was not related to oncogenic transformation of the transplanted cells, but rather due to massive BM aplasia and multi-organ lesions of infiltrating myeloid cells, a phenotype highly reminiscent of the most aggressive forms of human mixed histiocytoses[18]. Given that in our model mouse survival does not depend on human hematopoiesis but rather on the recovery of the murine hematopoiesis after sublethal irradiation, we report that the observed lethality can be ascribed to the detrimental

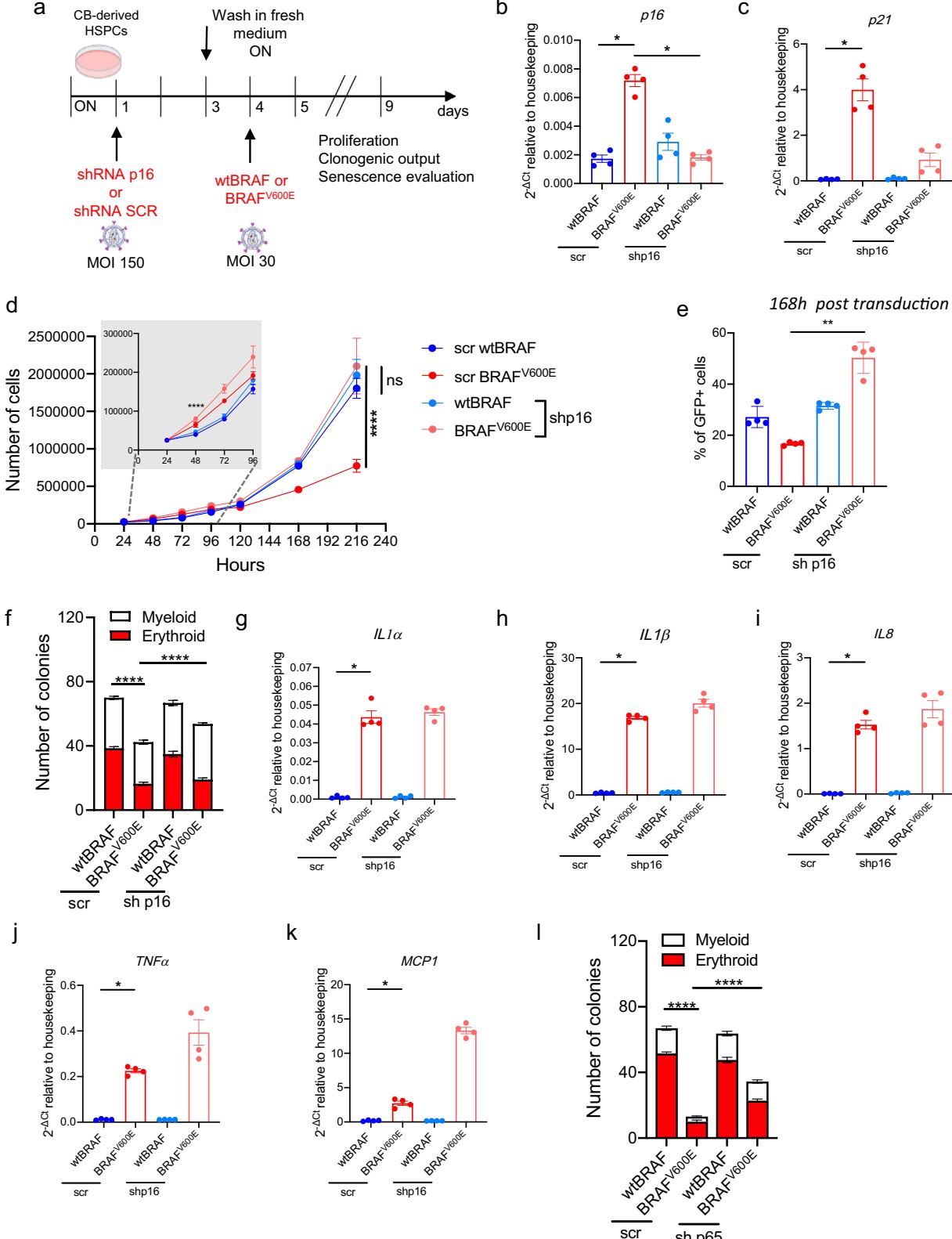

effect of oncogene-expressing human HSPCs on murine hematopoiesis and more in general on mouse health.

Notably, BRAF[V600E] expression in HSPCs resulted in a marked myeloid skewing of both human and murine hematopoiesis, a condition described in several chronic inflammatory diseases, viral infections, and during organismal aging[9,19–21]. The observed myeloid skewing in the human graft was the result of a strong

decline in the output of lymphoid and erythroid hematopoietic lineages. In support of this view, our comprehensive immunophenotypical characterization of the BM composition in the early phases of hematopoietic reconstitution revealed that oncogene activation led to a strong impairment in the pool of primitive hematopoietic stem cells, multipotent lymphoid and erythroid progenitors, while sparing more committed myeloid cells.

**Fig. 6 Dissection of the contribution of cell cycle arrest and SASP in BRAF$^{V600E}$-expressing HSPCs. a** Experimental strategy adopted to dissect p16 and SASP induction. Relative mRNA expression of (**b**) p16 and (**c**) p21 upon treatment with scrambled (scr) shRNA or shRNA against p16/ARF (shp16). Data are presented as mean $+/-$ SEM. Each dot represents an independent donor ($n = 4$). Statistical test: Kruskal−Wallis with Dunn's multiple comparison. **d** Growth curves of wtBRAF or BRAF$^{V600E}$-expressing HSPCs depleted or not for p16. $n = 4$ independent donors. Data are presented as mean $+/-$ SD. Statistical test: Kruskal-Wallis with Dunn's multiple comparison at 48 and 216 h. Inset indicates growth curves at early time points. **e** Percentage of GFP$^+$ cells of wtBRAF or BRAF$^{V600E}$-expressing HSPCs depleted or not for p16. Data are presented as mean $+/-$ SD. Each dot represents an independent donor ($n = 4$). Statistical test: Kruskal−Wallis with Dunn's multiple comparison. **f** Clonogenic potential upon treatment with scr or shp16, in methylcellulose assays at 120 h post transduction, wtBRAF: $n = 12$ repeated measurements for four independent donors; BRAF$^{V600E}$: $n = 12$ repeated measurements for four independent donors. Statistical test: Two-tailed Mann−Whitney. **g−k** Gene expression analysis of SASP cytokines (*IL1α, IL1β, IL8, TNFα,* and *MCP1*) in wtBRAF or BRAF$^{V600E}$-expressing cells upon treatment with scr or shp16. Data are presented as mean $+/-$ SEM. Each dot represents an independent donor transduced with BRAF$^{V600E}$ ($N = 4$). Statistical test: Two-tailed Mann−Whitney. **l** Clonogenic potential of wtBRAF and BRAF$^{V600E}$-expressing HSPCs upon treatment with scrambled shRNA (scr) or shRNA against RELA/p65 (shp65) in methylcellulose assays at 120 h post transduction with wtBRAF or BRAF$^{V600E}$ vectors. $n = 12$ repeated measurements for four independent donors in each group. Data are presented as mean $+/-$ SEM. Statistical test: Two-tailed Mann−Whitney. All graphs in this figure display data from: wtBRAF treated with scr in dark blue or with shRNA either against p16 in light blue; BRAF$^{V600E}$ treated with scr in dark red or with shRNA either against p16 in light red. Gene expression values are represented as $2^{-\Delta Ct}$ relative to housekeeping gene. \**p* < 0.05; \*\**p* < 0.01; \*\*\*\**p* < 0.0001.

Differences in transduction levels could not explain the observed phenotype as lymphoid progenitors had less vector marking than myeloid counterparts. Instead, the impairment in lymphoid output that we report in our humanized model could be explained by an increased sensitivity to oncogenic stress of this specific cell type, in agreement with the decreased survival of lymphoid murine progenitors reported in response to genotoxic stress[22]. However, we cannot completely rule out the possibility that a myeloid specification bias can occur upon oncogene activation and be linked to a metabolic and transcriptional reprogramming of hematopoietic progenitors towards the myeloid lineage as described in other settings[19,23].

Interestingly, in the late phases of hematopoietic reconstitution, we found that the detrimental impact of BRAF$^{V600E}$ activation on lymphoid cells was not only confined to cells expressing the activated oncogene but also involved bystander cells, indicating a strong non-cell-autonomous effect. At the basis of this phenomenon, we found that oncogene expression in HSPCs, after an initial burst of proliferation, led to the appearance of oncogene-induced senescence features, including induction of cell-cycle inhibitors (p16 and p21), increased SA-β-Gal activity, DNA-damage response activation, and SASP induction, which led to increased levels of a plethora of pro-inflammatory cytokines. In turn, senescence establishment results in reduced clonogenic potential of oncogene-expressing HSPCs. To disentangle the contribution of senescence, SASP-related inflammation, and cell cycle arrest to the observed clonogenic impairment of BRAF$^{V600E}$-expressing HSPCs we depleted p16/ARF or blunted NFkB activity prior to oncogene expression and found that, while p16 depletion only modestly rescued proliferation block and HSPC clonogenic potential without affecting the SASP, NFkB depletion and consequent SASP suppression resulted in greater recovery of colony-forming capacity of BRAF$^{V600E}$-expressing cells, highlighting the key role of SASP factors in HSPC dysfunctions. Of note, when we assessed p16 and SA-β-Gal levels in the myeloid progeny of bone marrows of mice transplanted with BRAF$^{V600E}$-expressing HSPCs we detected senescence in both oncogene and non-oncogene-expressing cells. In agreement with case reports from histiocytic patients describing p16-positive cells in lesions[24], our own immunohistochemical analyses on patient samples showed p16 activation, SA-β-Gal activity, and TNFα accumulation in mutated BRAF$^{V600E}$-positive, as well as in bystander cells. We show that SASP is key for the transmission of senescence features to non-oncogene-expressing HSPCs as testified by the transfer of conditioned media enriched for factors secreted by oncogene expressing HSPCs. Thus, our findings provide a molecular explanation on why in histiocytic lesions

where only a minor fraction of histiocytes carries the genetic alteration, the non-mutated cells within the same lesion are phenotypically indistinguishable from mutated histiocytes. Given these results, we propose a model in which BRAF$^{V600E}$-expressing histiocytes can seed virtually in every tissue, secrete massive amounts of SASP factors, which contribute to the recruitment of circulating leukocytes in the lesions and at the same time turn them into senescent histiocytes. This phenomenon establishes a self-sustaining vicious cycle where senescence-induced rampant inflammation triggers a potentially lethal condition itself. Along this model, clinical evidence suggests that human lesions progress due to the continuous recruitment of healthy and mutated monocytes constantly produced by the BM. Importantly, in our humanized mouse model senescence establishment precedes the observed lymphoid impairment and disease onset, further supporting its causative role in disease pathogenesis.

To identify the soluble factors key for the transmission of the paracrine senescence phenotype in the hematopoietic system, we assessed the role of IL1α, IL1β, and TNFα, important mediators of stress-induced inflammatory programs and upstream regulators of SASP. To our surprise, chronic exposure of human HSPCs to IL1β had no impact on HSPCs proliferation and senescence establishment while IL1α had only a modest effect even at the highest dose tested. Instead, our functional studies identify TNFα as the upstream inducer of SASP cytokines such as IL1α, IL1β, MCP1, and IL8 and senescence-associated cell cycle inhibitors. Senescence establishment upon chronic TNFα exposure was preceded by an initial wave of proliferation, thus mimicking the effects observed early upon oncogene activation in HSPCs. This observation is consistent with a previous study reporting that acute short-term exposure of murine primitive HSCs to TNFα exerts pro-survival and mitogenic effects[25]. However, this increased proliferation was transient and eventually followed by cell cycle arrest and accumulation of senescence-associated markers. Our model supports the notion that chronic TNFα production plays a major role in the pathogenesis of inflammatory neoplasms as its pharmacological blockade improves mice survival, ameliorates disease phenotype, abrogates paracrine senescence, and rescues lymphoid and erythroid outputs, especially in bystander non-oncogene-expressing cells. Of note, mice from the BRAF$^{V600E}$ group still died of bone marrow failure upon TNFα blockade; this phenotype could be explained by the fact that TNFα blockade treatment did not eradicate BRAF$^{V600E}$-expressing senescent cells from the lesions but rather limited the detrimental effects of SASP on bystander cells. Nonetheless, our data provide a mechanistic explanation of the therapeutic benefits reported for TNFα blockade in the treatment of milder forms of histiocytoses[26].

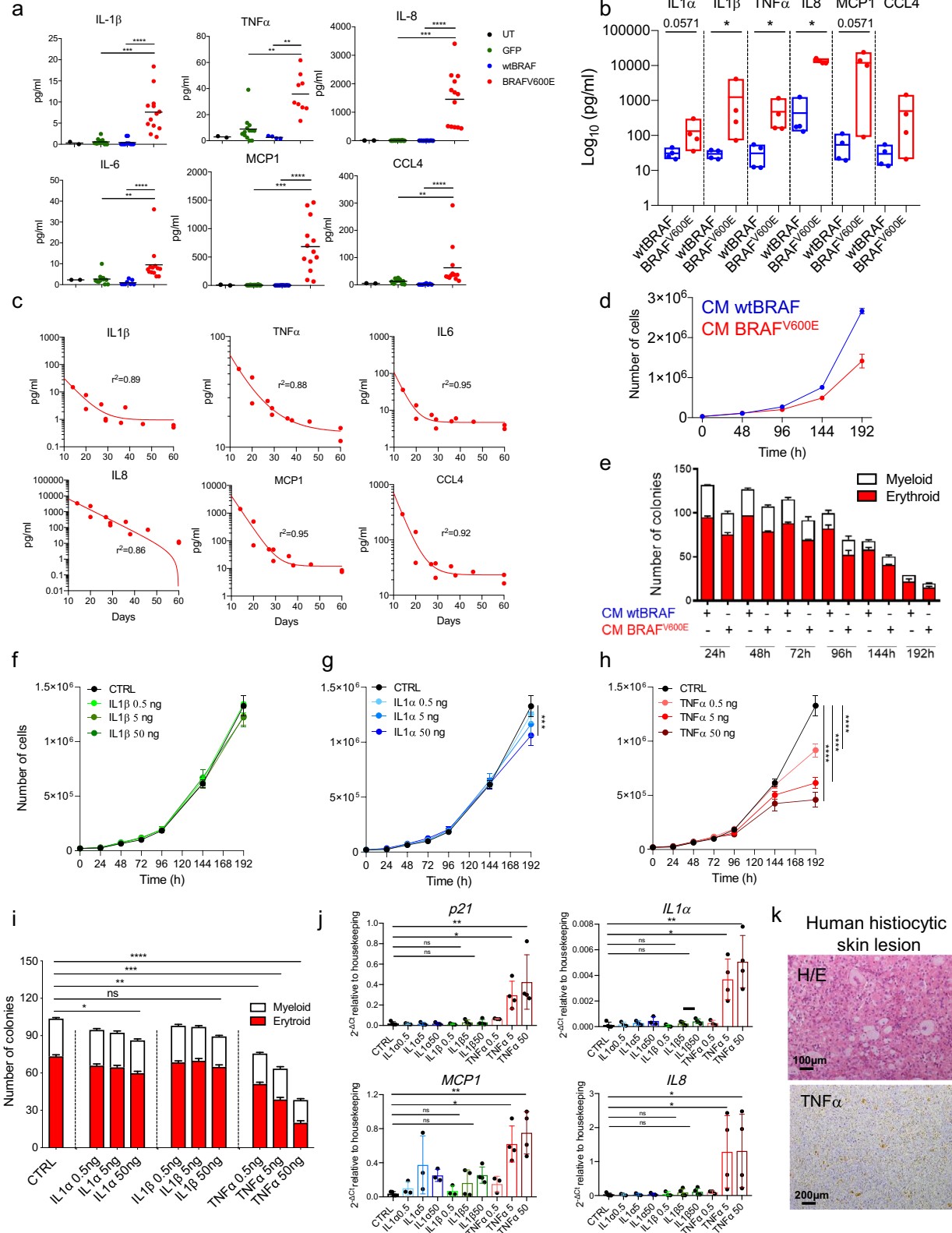

Currently, the clinical management for histiocytoses remains suboptimal, as it usually involves highly toxic chemotherapy and steroids regimens, type-I interferon, or targeted therapies, which specifically block MAPK pathway signaling[27]; however, these treatments are associated to significant side effects and do not provide a definitive cure for patients. Despite the intrinsic limitation of human mouse hematochimeras to reproduce the complexity of physiologic interactions between hematopoietic cells and tissue microenvironment, our humanized mouse model represents a unique and powerful tool for testing innovative senescence-based therapeutic interventions. In addition to SASP inhibition, selective eradication of senescent cells by senolytic drugs[28] might be beneficial, contribute to reduce tissue dysfunction and inflammation, and provide a definitive cure for patients.

**Fig. 7 SASP factors from senescent HSPCs induce paracrine senescence in non-oncogene-expressing cells.** Concentrations of human pro-inflammatory factors in (**a**) blood serum of mice at euthanasia or (**b**) in cultured media 120 h post-transduction. For (**a**), each dot represents a mouse: untreated (UT, black, $n = 2$), GFP (green, $n = 13$), wtBRAF (blue, $n = 4$), BRAF$^{V600E}$ (red, $n = 13$). For (**b**), wtBRAF (blue), and BRAF$^{V600E}$ (red), four donors for each group. Data are presented as box plots minimum to maximum with lines representing means. **c** Correlation between blood cytokine levels and survival of BRAF$^{V600E}$ mice ($n = 11$). Non-linear regression one-phase decay. The goodness of fit: $R^2$. **d** Growth curve of HSPCs exposed to CM from wtBRAF or BRAF$^{V600E}$ HSPCs. $n = 9$ for three donors/group. Data are presented as mean values +/− SD. **e** Clonogenic potential of HSPCs exposed to CM from wtBRAF or BRAF$^{V600E}$ HSPCs. $n = 7$ for three donors/group. Data are presented as mean values +/− SEM. **f–h** Growth curves of HSPCs exposed to cytokines: (**f**) IL1α (light green: 0.5 ng/ml, medium green: 5 ng/ml and dark green: 50 ng/ml), (**g**) IL1β (light blue: 0.5 ng/ml, medium blue: 5 ng/ml and dark blue: 50 ng/ml) and (**h**) TNFα (light red: 0.5 ng/ml, medium red: 5 ng/ml and dark red: 50 ng/ml). Technical replicates: CTRL ($n = 5$), IL1α-IL1β-TNFα ($n = 4$). CTRL values (black) are repeated in **f**, **g**, and **h**. Data are presented as mean values +/− SD. Statistical test: linear mixed-effects model (LME), followed by post hoc analysis at 24 and 192 h. See also Source Data file. **i** Clonogenic potential of HSPCs treated with recombinant cytokines for 96 h. CTRL($n = 4$), IL1α ($n = 4$), IL1β ($n = 4$), TNFα ($n = 4$). Data are presented as mean values +/− SEM. **j** Gene expression analysis of $p21$, IL1α, MCP1, and IL8 upon cytokine treatments (three donors for: CTRL, IL1α0.5, IL1α5, IL1α50, IL1β0.5, and TNFα0.5; four donors for: IL1β5, IL1β50, TNFα5, and TNFα50). Data are presented as mean +/− SD. **k** Representative images of a skin lesion from an histiocytosis patient stained for TNFα. Tissue morphology analyzed by hematoxylin and eosin (H/E). Scale bars: 100 or 200 µm. Statistical tests: for (**b**), two-tailed Mann−Whitney; for **a**, **i**, **j**, Kruskal−Wallis with post-Dunn's multiple comparisons. ns non significant; *$p < 0.05$; **$p < 0.01$; ***$p < 0.001$; ****$p < 0.0001$.

Altogether, we provide compelling experimental evidence that senescence establishment in human hematopoiesis links onco-genic mutations to the reactive inflammatory phenotype of hematological neoplasms. More broadly, our findings may contribute to the mechanistic understanding of hematopoietic dysfunctions observed in response to stress, inflammatory conditions, as well as during physiological aging.

## Methods

**Lentiviral vector constructs and production.** cDNA sequences for wtBRAF and BRAF$^{V600E}$ were obtained from Addgene (cat. #40775 and cat. #17544 respectively). cDNA sequences were amplified by PCR with oligonucleotide primers containing XmaI and SalI restriction sites (FW: agttaggcccgggcttaagga-tatcgccaccatggcggcgctgagcggtgg; REV: tgaaagcgtcgacctgcagaccggttcagtgga-caggaaacgcac). The PCR products were then digested with XmaI and SalI (New England Biolabs, Ipswich, MA) and ligated (T4 ligase New England Biolabs, Ips-wich, MA) into the pCCL.sin.cPPT.SV40polyA.eGFP.minCMV.hPGK.del-taLNGFR.Wpre vector devoid of Δ-LNGFR in order to express our transgenes under the human PGK promoter together with GFP as a marker gene. Plasmids for shRNA-encoding LVs were obtained from TRC Lentiviral shRNA Gene sets for $p16^{ink4A}/p19ARF$ and RELA/p65 (Horizon Discovery). VSV.G-pseudotyped third-generation SIN LVs were produced by calcium phosphate transient transfection into 293T cells as in[29].

**Culture conditions and transduction protocol.** CD34$^+$ HSPCs were isolated from human cord blood (CB) according to the TIGET09 protocol (approved by Ospe-dale San Raffaele Bioethical Committee) or purchased from Lonza. Transduction was achieved as describe previously[30]. Briefly, CD34$^+$ HSPCs were put in culture at the concentration of $5*10^5$ CD34$^+$ cells/ml in serum-free StemSpan medium (StemCell Technologies) supplemented with penicillin, streptomycin, glutamine, and human early-acting cytokines (SCF 100 ng/ml, Flt3-L 100 ng/ml, TPO 20 ng/ml, and IL6 20 ng/ml; all from Peprotech). Twenty-four hour after thawing, LVs were added at a MOI ranging from 1 to 30. Human recombinant cytokines IL1α (200LA), IL1β (201LB), and TNFα cytokines (210TA), purchased from R&D, were added at 5/10/50 ng/ml as indicated. Cytokines were refreshed every 48 h by removing 30% of the total volume and replacing with fresh media. Conditioned media (CM), retrieved from HSPCs transduced with either wtBRAF or BRAF$^{V600E}$ LVs, was applied to fresh CB-derived CD34$^+$ at a 1:1 ratio with complete StemSpan medium (StemCell) with or without TNFα inhibitor (Infliximab, IBL America) at 50 ng/µl. After every 48 h, the medium was refreshed by adding a new mix of StemSpan/CM/inhibitors medium to HSPCs. Cells were kept at 37 °C in a 5% CO$_2$ water jacket incubator (Thermo Scientific). CFU-C assay was performed by plating 800 cells in a semi-solid medium (MethoCult H4434, StemCell Technologies) supplemented with penicillin and streptomycin, at different hours post transduc-tion as indicated. Two weeks after plating, colonies were counted and classified as myeloid (white) or erythroid (red) according to morphological criteria.

**Flow cytometry.** For immunophenotypic analyses (performed on FACSCanto II; BD PharMingen), single-stained and Fluorescence minus one stained cells were used as controls. 7-AAD (Biolegend) was included in the sample preparation for flow cytometry according to the manufacturer's instructions to exclude dead cells from the analysis. Cell sorting was performed using MoFlo XDP Cell Sorter (Beckman Coulter) or FACS Aria Fusion (BD Biosciences). WBD protocol was performed according to ref. [15]. Briefly, after RBC lysis, the samples were incubated with mouse purified Rat anti-mouse CD16/CD32 (mouse BD Fc block™) solution (BD Bioscience) for 10 minutes at RT, to avoid unspecific binding to murine cells. After Fc blocking, cells were labeled with fluorescent antibodies against human

antigens CD3 (1:50), CD56 (1:50), CD14 (1:50), CD41/61 (1:50), CD13 (1:50), CD34 (1:20 or 1:50), CD45RA (1:20) and CD33 (1:50), CD66b (1:50), CD38 (1:33), CD45 (1:33 or 1:50), CD90 (1:33), CD10 (1:20), CD11c (1:20), CD19 (1:50), CD7 (1:50), and CD71 (1:20) or murine antigens CD45 (1:100), CD11b (1:100), and Gr1 (1:100). After surface marking, the cells were incubated with PI (Biolegend) to stain dead cells. All samples were acquired through BD Symphony A5 (BD Bioscience) cytofluorimeter after Rainbow beads (Spherotech) calibration and raw data were collected through DIVA software (BD Biosciences). For intracellular p16 staining, HSPCs collected from BM were fixed with PFA 4% for 20 min at RT, then per-meabilized with saponine-based solution for 15 min and incubated with primary antibody against p16 (1:100) for 1 h. After 40 min of incubation with a fluores-cently labeled secondary antibody (Donkey anti-Mouse IgG, Alexa Fluor 647, 1:1000), cells were washed with DPBS and p16 signal was recorded through BD Canto (BD Bioscience) cytofluorimeter after Rainbow beads (Spherotech) cali-bration and raw data were collected through DIVA software (BD Biosciences). Apoptosis analysis was performed by 7-AAD staining in combination with Annexin V (BioLegend) according to the manufacturer's instructions. The data were subsequently analyzed with FlowJo software Version 10.5.3 (BD Biosciences) and the graphical output was generated through Prism 8.3.0 (GraphPad software). TSNE plots were generated through FlowJo software plug-in. Gating strategies are shown in Supplementary Figs. 4, 5a, 6k, 7b, 10, and 11.

**SA-β-galactosidase assay.** Induction of senescence was assessed by staining transduced HSPCs with a senescence β-galactosidase (SA-β-Gal) Staining Kit (Cell Signaling Technology) according to manufacturer's instructions[31]. In detail, mul-titest slides were treated for 20 min with poly-L-lysine solution (Sigma-Aldrich) at 1 mg/mL concentration. After two washes with DPBS solution, cells were seeded on covers for 20 min and fixed with PFA 4% for 20 min at RT and then incubated overnight with the SA-β-Gal staining solution (pH = 6.0) to reveal SA-β-Gal activity. Nuclei were then stained with DAPI (Sigma-Aldrich), and images were acquired with a Nikon Eclipse inverted microscope. At least 100 nuclei per sample were used for the quantification of SA-β-Gal-positive cells. For assessment of SA-β-Gal activity on tissue sections, which appears as bright-blue granular staining in the cytoplasm of cells, frozen sections (3 µm) of a Erdheim-Chester (ECD) xanthe-lasma excised for diagnostic purposes were stained according to the manufacturer's instructions (Cell Signaling Technology). For FACS-based detection of senescent cells, staining of BM purified HSPCs was performed using a fluorescence senes-cence β-galactosidase (Spider-β-Gal) Staining solution (Dojindo), according to manufacturer's instructions with minor modifications to the protocol. In detail, HSPCs were plated in a 96 multiwells plate at the concentration of 1 million of cells/mL in supplemented Stemspan as above, then a Chloroquine solution was added to a final concentration of 150 µM/mL. After 1 h incubation, Spider-β-Gal Staining solution was added (final concentration 1 µM) and incubated for 15 min. Finally, stained cells were washed with DPBS solution, and then the fluorescence signal was recorded through BD Canto (BD Bioscience) cytofluorimeter after Rainbow beads (Spherotech) calibration, and raw data were collected through DIVA software (BD Biosciences). The data were subsequently analyzed with FlowJo software Version 10.5.3 (BD Biosciences) and the graphical output was generated through Prism 8.3.0 (GraphPad software).

**Luminex assay.** R&D system Luminex Kit was used to analyze cytokine's secretion both in sera obtained from peripheral blood of mice transplanted with wtBRAF and BRAF$^{V600E}$-expressing HSPCs and in supernatants derived from wtBRAF and BRAF$^{V600E}$-expressing HSPCs in liquid culture. Customized Luminex plates were obtained to screen for: IL1α, IL8, IL6, MCP1, IL1β, TNFα, IFNγ, IL12, GM-CSF, and CCL4. Assays were run as per manufacturers' instructions with standards and samples in technical duplicates. Data were acquired on a calibrated Bio-Plex MAGPIX multiple reader system (Bio-Rad) and visualized with Bio-Plex manager Software.

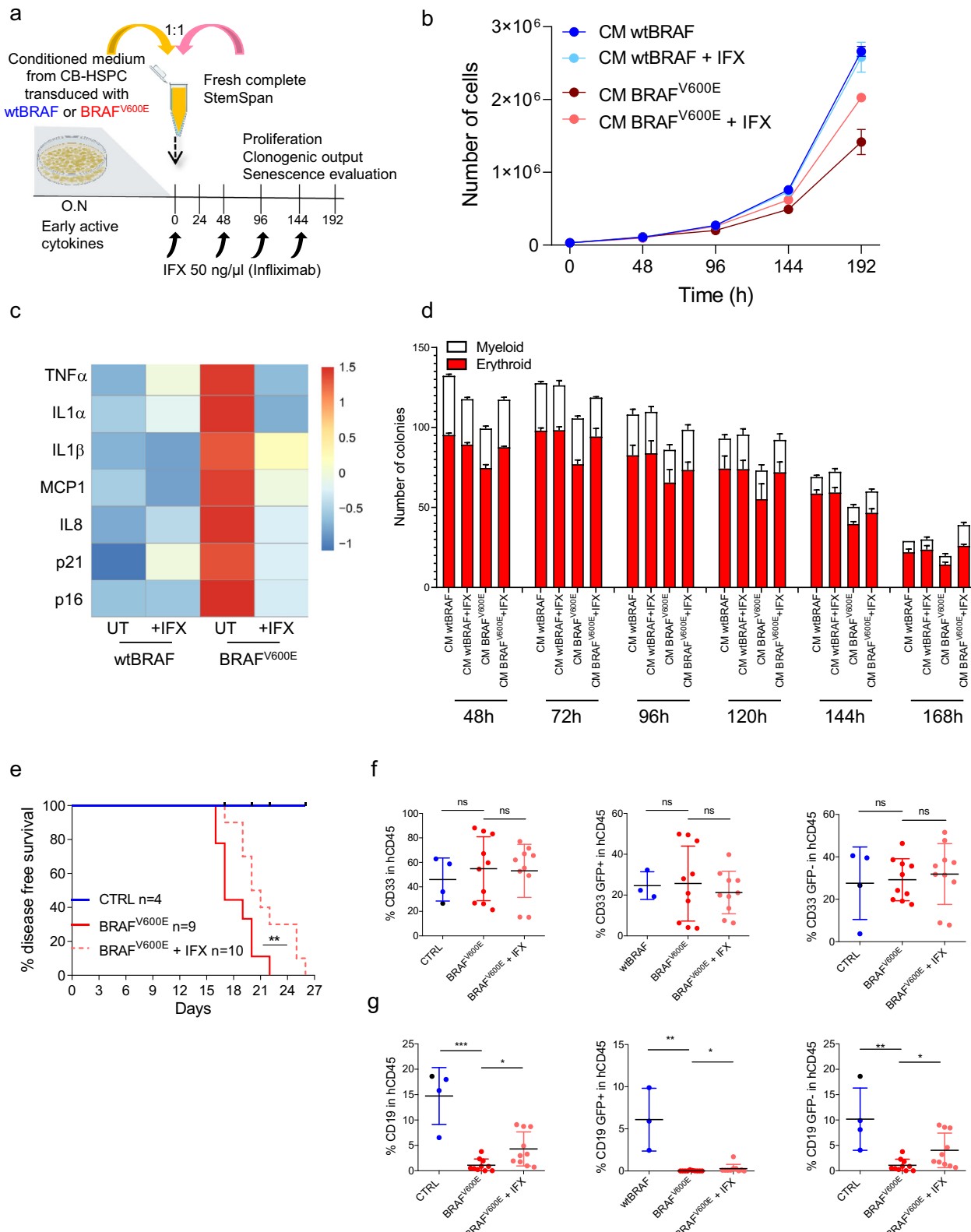

**Gene expression analysis**. Total RNA was isolated either using miRNeasy Micro Kit (QIAGEN) with DNase treatment through RNase-free DNase Set (QIAGEN) or using RNeasy Plus Micro Kit (QIAGEN), according to the manufacturer's instructions. cDNA was obtained with iScript cDNA Synthesis Kit (Bio-Rad) then pre-amplified with TaqMan PreAmp Master Mix (ThermoFisher). Finally, transcripts were quantified through q-PCR using Fast SYBR Green Master Mix (ThermoFisher) in a Viia7 Real-time PCR thermal cycler. The concentration of primers was determined through standard curve method

optimization to reach 100% amplification efficiency. Relative expression of analyzed genes was determined normalizing to *GUSB* housekeeping gene expression and then represented as $2^{\wedge-\Delta Ct}$. For primers sequences, see Supplementary Table 1 (RT-qPCR primer list).

**Total RNA sequencing library preparation and analysis**. Total RNA was isolated using RNeasy Plus Mini Kit (QIAGEN), according to the manufacturer's instructions. RNA was quantified with The Qubit 2.0 Fluorometer (ThermoFisher) and

**Fig. 8 TNFα inhibition rescues paracrine HSPC dysfunction upon oncogene activation. a** Schematic representation of cell treatments with conditioned medium derived from HSPCs transduced with wtBRAF or BRAF$^{V600E}$ lentiviral vectors, treated or not with a TNFα inhibitor (Infliximab; IFX) at the indicated concentration and time points. **b** Growth curve of HSPCs cultured with CM derived from HSPCs transduced with wtBRAF (dark blue) and BRAF$^{V600E}$ (dark red) treated or not with IFX (light blue for wtBRAF and light red for BRAF$^{V600E}$). $n = 9$ measurements for three donors in all groups. Untreated samples are same as in Fig. 7d. Data are presented as mean values $+/-$ SD. **c** Heatmap representation of gene expression analysis of senescence-associated markers in CM-treated HSPCs. Untreated $n = 4$, treated $n = 3$ biological replicates. **d** Clonogenic potential of HSPCs cultured with CM from transduced HSPC treated with IFX for 48-72-168-192 h: three measurements for $n = 1$ in all groups; 72−96 h: nine measurements for $n = 3$ independent donors in all groups. Untreated samples are same as in Fig. 7e. Data are presented as mean values $+/-$ SD. **e** Disease-free survival analysis of mice transplanted with BRAF$^{V600E}$ HSPCs upon treatment with IFX (CTRL = 4 mice, blue; BRAF$^{V600E}$ = 9 mice, red continuous line; BRAF$^{V600E}$ + IFX = 10 mice, red dashed line). Log-rank test between IFX treated and untreated mice from the BRAF$^{V600E}$ group. **$p < 0.01$. Immunophenotypical analysis of (**f**) myeloid output (CD33$^+$) or (**g**) lymphoid output (CD19$^+$) with GFP$^+$ and GFP$^-$ lineage for each experimental group. (CTRL = 4 mice, blue for wtBRAF and black for UT; BRAF$^{V600E}$ = 9 mice, red; BRAF$^{V600E}$ + IFX = 10 mice, light red). Data are presented as mean values $+/-$ SD. Statistical test: Kruskal−Wallis with Dunn's multiple comparisons comparing to BRAF$^{V600E}$ group. *$p < 0.05$;**$p < 0.01$; ***$p < 0.001$; ns non-significant.

quality was assessed by Agilent 4200 TapeStation (Agilent Technologies). Minimum quality was defined as RNA integrity number (RIN) > 7. 1 ng of total RNA was used for library preparation with SMART-Seq® v4 Ultra® Low Input RNA Kit (Takara Bio USA, Inc) and sequenced on a NextSeq 500 (Illumina). The quality of the reads was determined using FastQC and low-quality sequences were trimmed using trimmomatic. Reads were then aligned to the human reference genome (GRCh38/hg38) using STAR, with standard input parameters, and gene counts were produced using Subread featureCounts, using Genecode v31 as gene annotation. Transcript counts were processed using the R/Bioconductor package edgeR, normalizing for library size using trimmed mean of $M$-values, and correcting $p$-values using FDR. GSEA was performed considering different datasets (Gene Ontology, KEGG Pathway Database, Reactome Pathway Database, Molecular Signatures Database) using clusterProfiler (v 3.8.1, http://bioconductor.org/packages/release/bioc/html/clusterProfiler.html) by pre-ranking genes according to Log2FC values. Volcano plots have been used to display RNA-seq results plotting the statistical significance ($P$ value) versus the magnitude of change (fold change).

**Immunofluorescence analysis.** Glass slides (ten-well, MP Biomedicals) were coated with poly-L-lysine after incubation for 20 min with a 1 mg/ml concentrated solution (Sigma-Aldrich). Afterward, slides were covered with $0.5/1 \times 10^5$ cells for 20′ then fixed with a 4% paraformaldehyde solution (Santa Cruz Biotechnology) for other 20′. After permeabilization with 0.2% Triton X100, a 0.5% BSA and 0.2% fish gelatin solution in DPBS was applied to block aspecific interactions, then cells were incubated 1 h at RT with the following primary antibodies: p16 antibody (1:100), 53BP1 antibody, (1:600), Anti-phospho ATM (Ser1981) antibody, (1:100). Slides were then washed with DPBS and incubated for 1 h at RT with either of the secondary antibodies: Donkey anti-Mouse IgG, Alexa Fluor 647 or Donkey anti-Rabbit IgG, Alexa Fluor 568 diluted 1:1000. Nuclei were stained using DAPI (Sigma-Aldrich) and, finally, Aqua-Poly/Mount solution (Polysciences. Inc.) was used to mount coverslips (Bio-Optica). Fluorescent images were obtained using Leica SP2 and Leica SP5 Confocal microscopes, DDR foci quantification was carried out using ImageJ64 (version 1.47).

**CD34$^+$ HSPC xenotransplantation studies in NSG mice.** *NOD.Cg-Prkdc$^{scid}$ IL2ry$^{tm1Wjl}$/SzJ* (NSG) mice were purchased from The Jackson Laboratory (cat. #005557 Jackson Laboratories, Bar Harbor, ME) and were maintained in specific-pathogen-free conditions. The procedures involving animals were designed and performed with the approval of the Animal Care and Use Committee of the San Raffaele Hospital (IACUC #1131) and communicated to the Ministry of Health and local authorities according to Italian law. For transplantation, $1 \times 10^5$ CD34$^+$ cells 24 h post-transduction were injected intravenously into 8-weeks old female sublethally irradiated NSG mice (150−180 cGy). Mice were monitored three times a week by weight assessment and general appearance, taking into consideration motility, fur conditions, kyphosis, and other signs of disease. Human CD45$^+$ cell engraftment and the presence of transduced cells were monitored by a serial collection of blood from the mouse tail and at the time of euthanasia BM, spleen, and distal organs were harvested and analyzed. Mice were euthanized when they lost 20% of their body weight post-transplant.

**Immunohistochemical analyses.** Tissues have been fixed in neutral tamponed formalin 10% and included in paraffin. They have been cut in 4 μm sections with a Microm HM 355S (Thermo Fisher Scientific, Waltham, USA). Mouse monoclonal antibody VE1, was used to detect the BRAF$^{V600E}$ mutation. Stainings were performed with Bond-Max (Leica Biosystems). Antigen retrieval was performed during 60 min at 96 °C in pH9 buffer Bond Epitope Retrieval Solution 2 (Leica Biosystems). VE1 hybridoma supernatant was diluted one-third and incubated at 37 °C for 32 min. Staining was revealed with bond polymer refine red detection kit (Leica Biosystems).

The same procedure was carried out for the other commercial antibodies against: CD14, CD1a, CD68/PGM1, CD207, S100, p16, Ki67, and TNFα.

**Statistical analyses.** Data were expressed as means ± SEM or means ± SD in the graphical form of dot plots or box plots minimum to maximum with lines representing means. Linear and non-linear regression model were fitted to test for the presence of linear/nonlinear relationships. To compare effects between independent groups (also at a given time point), Mann−Whitney test (for two groups). or Kruskal−Wallis test followed by Dunn's multiple comparisons (for more than two groups) were performed. To test for the presence of linear/nonlinear relationships between survival, cytokine secretion, and myeloid skewing with the amount of transduced cells, non-linear regression models were performed. The log-rank test was used to assess the difference between groups with respect to disease-free survival.

A LME[31] model was employed to estimate the longitudinal trend of a number of cells (in log scale) or of the number of colonies and evaluate the differences among groups (BRAFV600E vs wtBRAF) for data in Figs. 4g, h and 7f–h). Both linear and quadratic terms for time were included in the mixed models to account for the nonlinear trajectories over time. LME was estimated in R (version 3.6.0) by means of the nlme package, while the emmeans package was used to perform the post hoc analysis and compute the pairwise comparisons of treatment groups at a fixed time point. p-values were adjusted using Bonferroni's correction. This procedure was applied to properly account for the dependence structure among observations, by including additional random-effect terms, thus considering in the model unobservable sources of heterogeneity among experimental units. Treatment group indicator and time variable, along with their interaction, were included as covariates in the model to identify potential differences in growth dynamics of treatment groups. Logarithmic transformation of the outcome was also considered to satisfy underlying model assumptions.

Analyses were performed using GraphPad Prism v8 and R statistical software. Differences were considered statistically significant at *$P < 0.05$, **$P < 0.01$, ***$P < 0.001$, ****$P < 0.0001$, "ns" stands for not significant.

**Reporting summary.** Further information on research design is available in the Nature Research Reporting Summary linked to this article.

## Data availability

All relevant data are included in the manuscript and its supplementary information files (also provided as a Source Data file), or from the corresponding authors upon request. The RNA-seq data from this study have been deposited in the NCBI Gene Expression Omnibus (GEO) under accession number: GSE144058. All software applications used were free or commercially available. Source data are provided with this paper.

## Code availability

Linear mixed-effects (LME) was estimated in R (version 3.6.0) by means of the nlme package and the emmeans package.

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

## Acknowledgements

We thank the Montini and Di Micco lab members for discussion, the San Raffaele Scientific Institute flow cytometry facility (FRACTAL), imaging facility (ALEMBIC). We thank L. Naldini from SR-Tiget for critical reading of the manuscript. We thank Federica Cugnata and Clelia Di Serio from CUSSB-University Center for Statistics in the Biomedical Sciences for the LME statistical models in Figs. 4g, h and 7f–h. We thank Barbara Camisa for technical help with mouse experiments and Amleto Fiocchi for technical help with the histology of mouse tissues. This work was supported by grants to EM from Telethon Foundation (TGT16B01 and TGT16B03) and to R. D. M. from Telethon (TGT16E05), a Career Development Award from Human Frontier Science Program (HFSP), an Advanced Research Grant from the European Hematology Association (EHA), a Hollis Brownstein Research Grant from Leukemia Research Foundation (LRF), the Interstellar Initiative on "Healthy Longevity" from New York Academy of Sciences (NYAS) and the Japan Agency for Medical Research and Development (AMED) and from the Italian Association for Cancer Research (AIRC under MFAG 2019—ID. 23321 project). A. C. was supported by an AIRC postdoctoral fellowship and is currently supported by the Lady Tata Memorial Trust International Award for Research in Leukemia 2020–2021. R. B., E. L., and D. G. conducted this study as partial fulfillment of their PhD in Molecular Medicine, International PhD School, Vita-Salute San Raffaele University, Milan, Italy. R. D. M. is a New York Stem Cell Foundation Robertson Investigator.

## Author contributions

R. B. and E. L. designed experiments, performed research, and interpreted data; K. G. and D. G. performed transplantation experiments and senescence analyses; P. G. and K. G. provided technical help with L. V. production; G. C., M. P., L. D., and C. D. performed pathology analyses and interpreted data. A. C., S. B., and I. M. performed library preparation and RNA seq bioinformatic analysis; D. C., M. N., and A. B. provided technical help with transplantation experiments; S. S. and L. B. C. performed WBD analysis under the supervision of A. A.; R. D. M. and E. M. conceived the study, interpreted data, supervised research, and wrote the paper.

## Competing interests

The authors declare no competing interests.
