## [Peer Review File · Nature Communications]

Reviewers' comments:

Reviewer #1 (Remarks to the Author); expert on oncogenic-induced senescence and blood cancer:

Biavasco-R,... ..Montini-E, Oncogene-induced senescence in hematopoietic progenitors features myeloid restricted hematopoiesis, chronic inflammation and histiocytosis
Submitted to Nature Communications

In this manuscript, Montini and colleagues address the role of oncogene-induced senescence (OIS) in mutant BRAF-driven histiocytosis, a myeloid malignancy characterized by activated macrophage-like cells and high-level inflammation. Using a mouse model reconstituted with stably BRAF-V600E-infected human hematopoietic stem and progenitor cells (HSPC), they observed a massive expansion of mononuclear myeloid cells, suppression of erythroid and lymphoid cells, and OIS with a strong pro-inflammatory senescence-associated secretory phenotype (SASP) that also, especially via TNF-alpha (TNFa) as the mediator, contributed to bystander (i.e. non-BRAF-mutant) cell senescence. The data indicate a – minor – impact of this TNFa-governed paracrine program on overall survival.

This is an interesting and technically well performed study that not only contributes to the pathogenic elucidation of a rare histiocytic disease, but sheds light on the cell-autonomous and non-cell-autonomous implications of BRAF-V600E-evoked senescence. Before publication in Nature Communications can be recommended, experimental approaches to or discussion of the following concerns are required.

Major concerns and comments

1. Previous work by other groups elucidated the BRAF-V600E moiety in syngeneic mouse models with respect to histiocytosis formation. The authors claim profound differences in OIS between men and mice as a reason why to study BRAF-V600E now in a human HSPC-propagated NSG model system. While transduction rates (as measured by the fraction of GFP-positive cells) differ from low-to-mid range percentages (thereby giving rise to non-transduced human hematopoiesis in recipient mice), the contribution of the immune-compromised remaining murine hematopoiesis has not been addressed. Is there any murine hematopoiesis recovering after sublethal irradiation during the observation period? And, vice versa, what would happen after 100% GFP+ flow-sorted HSPC transplanted into lethally irradiated recipients?
2. Fig. 4F: The accumulation/expansion of p16-high/Ki67-negative cells in BRAF-V600E-HSPC-propagated mice is difficult to understand. Although an anecdotal case in Fig. 4F, other lesions, including those in patients (Fig. 4K), seem to present similarly. How can cell numbers increase if cells don't proliferate? Are the authors claiming that susceptible cells get strongly attracted to the site of those senescence-like founder cells to settle there and undergo senescence by paracrine means? Or are those recruited cells already senescent when getting attracted to the seeding site?
3. Despite the massive induction of senescence at the HSPC level and their downstream differentiated cell offspring, and the strong pro-inflammatory aspect of the disease, the major threat still seems to be the macrophage-like monocytoid tissue invasion as a result of a

highly proliferative process. A genetic dissection of the SASP in general (not only at the level of TNF α), e.g. by blunting NF- κ B activity, and a genetic disruption of senescence, e.g. by depleting p53 or INK4a/ARF alleles, would have helped to disentangle senescence, SASP-like inflammation and proliferation.

4. Is the lethal histiocytic disease transplantable? If so, what does it take – HSPC isolated from diseased bone marrow, or just peripheral blood or cells from p16-high tissue lesions?

Minor concerns

1. Fig. 1F: It would have been more informative to see cells isolated from BRAF-V600E recipients being GFP+ and marker-positive (e.g. CD68, CD207, S100) by flow cytometry or immunofluorescence

2. Fig. 2: The control group is a mixed bag of untransduced, GFP-only and BRAF-wt-infected HSPC – did it matter, whether the cells were actually infected (i.e. potential myeloid bias due to preferred vector integration sites)? Suppl. Fig. 2D indicates that there is a clear difference between BRAF-wt vs. BRAF-V600E, however, at an extremely low relative extent of CD19+ B-cells (accounting for around 2% of the overall cellularity). Is the relative loss of both BRAF-wt- and BRAF-V600E-positive CD19+ B-cells (Fig. 2O) due to OIS, also in response to wild-type BRAF overexpression?

3. Fig. 2J is not displayed properly in my PDF

Reviewer #2 (Remarks to the Author); expert on haematopoiesis and inflammation:

This manuscript entitled “Oncogene-induced senescence in hematopoietic progenitors features myeloid restricted hematopoiesis, chronic inflammation and histiocytosis” reports that oncogene BRAF V600E induces a senescence-like phenotype in human hematopoietic progenitors contributes to the bone marrow aplasia and histiocytosis development in a humanized mouse model. This manuscript also reports that the SASP factors (senescence-associated secretory phenotype) secreted by oncogene-induced senescent cells also induce a senescence-like phenotype in non-oncogene expressing cells, therefore triggering a cascade of systematic inflammation. This study provides valuable information to understand how small numbers of oncogene-induced senescent cells can initiate a vicious cycle and contribute to the deterioration of health.

Altogether this is an interesting manuscript that is in general well done. To assist the authors in substantiating their conclusions we have provided a list of concerns that need to be addressed.

Major points:

1. The identity of the senescent cells from the bone marrow images in Figure 4 is not clear. p16 and beta-galactosidase can be induced in macrophages as part of a reversible response to physiological stimuli (PMCID 5611982) hence the result of these two markers in hematopoietic cells needs to be cautiously interpreted. The authors need to show whether these markers overlap with macrophages entirely or are also observed in non-macrophage

human hematopoietic cells. Likewise, the identity of the cells as human or mouse derived, V700E or WT is not clear and should be clarified either by imaging or flow cytometry to better support a model of in-trans induction of senescence. Along these lines, presumably senescence of progenitor cells is the trigger of the inflammatory phenotype based on the model proposed by the authors – to rigorously substantiate this, association of senescence markers with phenotypic HSPC in vivo should be provided.

2. This manuscript also showed that sporadic p16+ cells were detected in lung sections. However, as in point 1) it is unclear whether these cells were human CD45 cells or mouse CD45 cells, or mouse non-hematopoietic cells present in the tissue. Defining the origin species of these cells will strengthen this manuscript's central claim.

3. Histiocytosis is defined as an overabundance of tissue macrophages leading to pathology. The authors should provide more direct evidence for macrophage infiltration in their human and mouse tissue sections.

Minor points:

1. The authors performed transduction efficiency analysis and they might already have the data of transduction efficiency in different populations, HSC, MLP, PreBNK, CMP, GMP. These data should be shown as FACS plots for each population to allow the reader to gauge transduction efficiency and GFP expression levels. Adding this information will help to rule out the possibility that impaired lymphoid lineage is not caused by biased transduction efficiency in MLP and/or CMP populations.

2. It is stated in the manuscript that “Macroscopic analysis of the internal organs at euthanasia showed markedly paler long bones in mice from the BRAFV600E group, suggesting a profound alteration of BM cellularity, accompanied by a modest spleen enlargement (Figure 1E)”. A quantification for total BM cellularity will be needed to support this statement beyond the image of paler bones shown.

3. BRAF V600E induced necrosis was not ruled in or out in this study. Multiple studies have shown that BRAF mutations cause ulceration in melanoma patients (e.g., PMC4984864), an indication of tissue necrosis. Necrotic tissue triggers inflammation. Whether BRAF V600E overexpression in CD34+ cells induces necrosis is not addressed in this manuscript. Therefore, to draw a conclusion of oncogene induced senescence is the major driver for the phenotype described in this manuscript, the authors need to perform apoptosis and/or necrosis analysis on human CD34+ cells post BRAF V600E lentivirus transduction. Indeed, cell necrosis could also explain as an alternative hypothesis to TNF the trans-acting inflammatory phenotype.

“Oncogene-induced senescence in hematopoietic progenitors features myeloid restricted hematopoiesis, chronic inflammation and histiocytosis”
Nature Communications NCOMMS-20-03007-T

Reviewer #1 (Remarks to the Author); expert on oncogenic-induced senescence and blood cancer:

Biavasco-R,... ..Montini-E, Oncogene-induced senescence in hematopoietic progenitors features myeloid restricted hematopoiesis, chronic inflammation and histiocytosis
Submitted to Nature Communications

In this manuscript, Montini and colleagues address the role of oncogene-induced senescence (OIS) in mutant BRAF-driven histiocytosis, a myeloid malignancy characterized by activated macrophage-like cells and high-level inflammation. Using a mouse model reconstituted with stably BRAF-V600E-infected human hematopoietic stem and progenitor cells (HSPC), they observed a massive expansion of mononuclear myeloid cells, suppression of erythroid and lymphoid cells, and OIS with a strong pro-inflammatory senescence-associated secretory phenotype (SASP) that also, especially via TNF-alpha (TNFa) as the mediator, contributed to bystander (i.e. non-BRAF-mutant) cell senescence. The data indicate a – minor – impact of this TNFa-governed paracrine program on overall survival.

This is an interesting and technically well performed study that not only contributes to the pathogenic elucidation of a rare histiocytic disease, but sheds light on the cell-autonomous and non-cell-autonomous implications of BRAF-V600E-evoked senescence. Before publication in Nature Communications can be recommended, experimental approaches to or discussion of the following concerns are required.

We thank the Reviewer 1 for the appreciation of our study and for the constructive criticisms.

Major concerns and comments

1. Previous work by other groups elucidated the BRAF-V600E moiety in syngeneic mouse models with respect to histiocytosis formation. The authors claim profound differences in OIS between men and mice as a reason why to study BRAF-V600E now in a human HSPC-propagated NSG model system. While transduction rates (as measured by the fraction of GFP-positive cells) differ from low-to-mid range percentages (thereby giving rise to non-transduced human hematopoiesis in recipient mice), the contribution of the immune-compromised remaining murine hematopoiesis has not been addressed. Is there any murine hematopoiesis recovering after sublethal irradiation during the observation period?

We thank the Reviewer for raising this important point. To address this concern, we have performed a new set of experiments aimed to comprehensively address the impact of transplantation of BRAF^{V600E} expressing human HSPC on mouse hematopoiesis. A group of mice was transplanted with human HSPCs either expressing 30% of BRAF^{V600E}, or wtBRAF or left untreated, then euthanized at 24 days post transplantation for BM collection. We observed that in mice from the BRAF^{V600E} group, the total number of BM cells (by absolute quantification) was significantly lower than the one observed in the mice from control groups (New Figure 1F). The reduction in BM cellularity in BRAF^{V600E} group involved both murine and human CD45⁺ hematopoietic cells (New Supplementary Figure 1A, B) and could be only partially ascribed to the increased number of necrotic cells (by 7AAD positivity) in the BM compared to controls (New Supplementary Figure 1C). Interestingly, when we tested the relative percentage of the murine myeloid output in the BRAF^{V600E} group, we observed a marked skewing of the murine hematopoiesis towards the

monocytic/macrophagic lineage (CD11^{high}/Gr⁻/mCD45⁺) compared to the control groups (New Supplementary Figure 2C), while the granulocytic compartment (CD11^{high}/Gr⁺/mCD45⁺) did not appear to be affected in any of the groups analyzed (New Supplementary Figure 2D). In summary, both human and mouse hematopoietic cells in the bone marrow appear to be strongly reduced upon transplantation of BRAF^{V600E} expressing human HSPCs *in vivo*, with a marked myeloid skewing.

We have added these new results to the main text:

Page 6: "When focusing on the impact of oncogene-expressing human cells on murine hematopoiesis, the relative percentage of the murine myeloid output in the mice of the BRAF^{V600E} group compared to the other groups showed a marked skewing towards the monocytic/macrophagic CD11+high/Gr1-/mCD45+ lineage but did not involve granulocytes CD11+high/Gr1+/mCD45+ (Supplementary Figure 2C, D)."

And, vice versa, what would happen after 100% GFP+ flow-sorted HSPC transplanted into lethally irradiated recipients?

We thank the reviewer for this comment and acknowledge that it would be interesting to evaluate the hematopoietic output of oncogene-expressing human HSPCs upon complete eradication of the murine hematopoiesis by lethal irradiation. However, to our knowledge, if transplantation of human HSPCs takes place after lethal irradiation, mice will die early from bone marrow failure, regardless of if human HSPCs were expressing or not BRAF^{V600E}, because the survival of lethally irradiated NOD.Cg-Prkdc^{scid}Il2rg^{tm1Wjl}/SzJ immunocompromised mice (NSG) does not rely on the engraftment or activity of human hematopoietic cells but rather on the recovery of murine hematopoiesis. The above considerations render the experiment suggested on lethally-irradiated mice not feasible.

Nonetheless, to address this Reviewer point on the transplantation outcome of an enriched fraction of oncogene-expressing human cells, we would like to highlight some of our results obtained by transplanting variable fractions of BRAF^{V600E} expressing HSPCs (GFP+), reaching up to 76% of transduction level (New Figure 2E and New Supplementary Figure 2B, already in the first version of our manuscript). We found that mice transplanted with an high fraction of BRAF^{V600E} expressing HSPCs had a dramatic impairment of hematopoiesis and died significantly earlier compared to mice transplanted with low numbers of BRAF^{V600E} expressing HSPCs (New Figure 1D). Therefore, even though an experiment with lethally irradiated mice would not be feasible, based on these data, we can predict that transplanting 100% of BRAF^{V600E}- expressing HSPCs into sub-lethally irradiated mice would result in a rapid BM failure before the dissemination of myeloid cells into distal organs and before the onset of histiocytic lesions, jeopardizing the validity of our transplantation model for the study of the pathogenesis of human histiocytosis.

We have added these considerations to our discussion:

Page 16: "Given that in our model mouse survival does not depend on human hematopoiesis but rather on recovery of the murine hematopoiesis after sublethal irradiation, we report that the observed lethality can be ascribed to the detrimental effect of oncogene-expressing human HSPCs on murine hematopoiesis and more in general on mouse health."

2. Fig. 4F: The accumulation/expansion of p16-high/Ki67-negative cells in BRAF-V600E-HSPC-propagated mice is difficult to understand. Although an anecdotal case in Fig. 4F, other lesions, including those in patients (Fig. 4K), seem to present similarly. How can cell numbers increase if cells don't proliferate? Are the authors claiming that susceptible cells get strongly attracted to the site of those senescence-like founder cells to settle there and undergo senescence by paracrine means? Or are those recruited cells already senescent when getting attracted to the seeding site?

We thank the Reviewer for this comment and would like to better guide the Reviewer through our understanding of lesion formation, dissemination and growth. We show that BRAF^{V600E} expression in CD34⁺ HSPCs activated in culture elicits cell cycle arrest, which however is preceded by a phase of increased proliferation (New Figure 4G), as previously reported in human fibroblasts (PMID: 17136094, PMID: 17136093). Thus, at least in vitro, we report that oncogene-expressing HSPCs do expand initially, but they cease proliferating after accumulation of DNA damage, the main driver of the senescence phenotype. Similarly, in the bone marrow of mice transplanted with BRAF^{V600E} expressing HSPCs, we were able to report an expansion of myeloid progenitors (CD34⁺/CD33⁺ cells), especially in the GFP negative fraction, 2 weeks post transplantation at the time of sacrifice (New Supplementary Figure 2G), suggesting that some proliferation of oncogene expressing cells *in vivo* in the bone marrow may precede or accompany the observed myeloid-restricted hematopoiesis. Another important consideration in support of the initial proliferation of oncogene-expressing HSPCs is that we were able to recover by absolute count around 300.000 cells *per femur* three weeks upon transplantation of 100.000 BRAF^{V600E}-expressing HSPCs (New Supplementary Figure 1B). When focusing on histiocytic lesions, in our time course experiment (New Figure 4J) we reported that BRAF^{V600E} expressing cells with macrophagic morphology can be detected in small clusters as early as 9 days post transplantation and the lesions appeared increased in size already by days 15-24 post-transplant, with some ongoing proliferation (Ki67 positivity) in non-senescent cells (p16 negative) (New Figure 4J). Based on this experimental evidence, we speculate that during the early phases of hematopoietic reconstitution, BRAF^{V600E} expressing myeloid cells, after exiting the bone marrow, seed in the different organs and start the formation of the histiocytic lesion that become senescent *in loco* and recruits, likely via SASP factors, other circulating myeloid cells regardless of their oncogene expression (as also shown in New Figure 4I, where both GFP⁺ and GFP⁻ cells were detected within the lesion). Non-senescent myeloid cells may then become senescent at the seeding site by paracrine means. Lesions from patients with histiocytosis may increase in size over time with a similar mechanism.

We have added these considerations to our discussion:

Page 17: “Given these results, we propose a model in which BRAF^{V600E}-expressing histiocytes can seed virtually in every tissue, secrete massive amounts of SASP factors, which contribute to the recruitment of circulating leukocytes in the lesions and at the same time turn them into senescent histiocytes. This phenomenon establishes a self-sustaining vicious cycle where senescence-induced rampant inflammation triggers a potentially lethal condition itself. Along this model, clinical evidence suggests that human lesions progress due to continuous recruitment of healthy and mutated monocytes constantly produced by the BM”

As future perspective, following this Reviewer’s comment, we will attempt to implement our transplantation model with a luciferase-based BRAF^{V600E} expressing lentiviral vector to track oncogene-expressing cells upon transplantation by live-imaging and to perform *in vivo* evaluation of cell proliferation by Edu/BrdU incorporation. We are also optimizing a flow cytometry assay using C12FDG, cleaved by β -galactosidase producing a fluorescent product to reliable process dozen of mice for quantification of accumulated senescent cells within tissues.

3. Despite the massive induction of senescence at the HSPC level and their downstream differentiated cell offspring, and the strong pro-inflammatory aspect of the disease, the major threat still seems to be the macrophage-like monocytoid tissue invasion as a result of a highly proliferative process. A genetic dissection of the SASP in general (not only at the level of TNFa), e.g. by blunting NF-kB activity, and a genetic disruption of senescence, e.g. by depleting p53 or INK4a/ARF alleles, would have helped to disentangle senescence, SASP-like inflammation and proliferation.

To dissect the contribution of cell cycle arrest and SASP production to the observed phenotype of senescence induction in $BRAF^{V600E}$ expressing HSPCs, we took advantage of RNA interference to first downregulate the levels of p16, then transduced the HSPC with wtBRAF or $BRAF^{V600E}$ expressing lentiviral vectors and evaluated the clonogenic output, proliferation and senescence markers (New Figure 5A). In these experimental settings we observed that p16 inhibition resulted in reduction of cell cycle inhibitors (New Figure 5 B-C), rescued the proliferation block of $BRAF^{V600E}$ HSPCs (New Figure 5 D, E), and modestly impacted on the colony forming capability of oncogene expressing HSPCs (New Figure 5F). Interestingly, p16 inhibition did not affect $BRAF^{V600E}$ -induced SASP factors, as previously reported (PMID: 21880712), thus providing a mechanistic explanation of the modest increase in clonogenic potential of oncogene expressing cells (New Figure 5G-K). Conversely, when we inhibited p65, the subunit of the NF κ B heterodimer, we could not only dampen SASP but also alleviate the oncogene-induced cell cycle arrest, resulting in a stronger rescue of the clonogenic capacity of $BRAF^{V600E}$ expressing cells (see below for Reviewer only a merged version of New Figure 5F and 5L, with indication of fold changes related to the $BRAF^{V600E}$ group).

We have added the results obtained with this new analysis in the main text:

Page 12: “To dissect the contribution of p16-mediated cell cycle arrest to the observed phenotype of senescence induction in $BRAF^{V600E}$ expressing HSPCs, we first downregulated the levels of p16 by RNA interference, then transduced HSPCs with wtBRAF or $BRAF^{V600E}$ expressing vectors and evaluated their clonogenic output, proliferation and senescence emergence (Figure 5A). In this experimental setting we observed that p16 inhibition resulted in reduction of both cell cycle inhibitors p16 and p21 (Figure 5B, C), rescued the proliferation block of $BRAF^{V600E}$ HSPCs (Figure 5D, E), and partially rescued colony forming capability but did not alter the production of SASP cytokines, as previously reported (Figure 5F-K).

We then asked if we could dampen SASP cytokines by knocking down the master transcriptional regulator of pro-inflammatory cytokines NF- κ B. To test this hypothesis, we first downregulated the levels of RELA/p65, an essential subunit of the NF- κ B heterodimer, by RNA interference, then transduced HSPCs with wtBRAF or $BRAF^{V600E}$ expressing vectors. When we evaluated clonogenic

output and expression levels of inflammatory cytokines as well as cell cycle regulators, we observed a strong rescue of the clonogenic potential of oncogene-expressing HSPCs (Figure 5L), reinforcing the idea that the clonogenic impairment observed in in vitro cultured wtBRAF and BRAF^{V600E} transduced HSPCs (Figure 4H), is aided by the production of SASP cytokines. In agreement with this, we also report a consistent trend of decrease in several SASP cytokines mRNA levels upon RELA/p65 downregulation, and a reduction in p21 levels while no significant changes in p16 expression levels were observed (Supplementary Figure 4O). These data indicate that NFkB controls senescence establishment in oncogene-expressing HSPCs and that SASP suppression ameliorates HSPC clonogenicity.”

These data help us to disentangle two main senescence features (e.g. cell cycle arrest and SASP) and their role on hematopoietic dysfunction. We have also added these considerations to our discussion:

Page 17: “In turn, senescence establishment results in reduced clonogenic potential of oncogene-expressing HSPCs. To disentangle the contribution of senescence, SASP-related inflammation and cell cycle arrest to the observed clonogenic impairment of BRAFV600E-expressing HSPCs we depleted p16 or blunted NFkB activity prior to oncogene expression and found that, while p16 depletion only modestly rescued proliferation block and HSPC clonogenic potential without affecting the SASP, NFkB depletion and consequent SASP suppression resulted in greater recovery of colony forming capacity of BRAFV600E expressing cells, highlighting the key role of SASP factors in HSPC dysfunctions. Of note, when we assessed p16 and SA-β-Gal level in the myeloid progeny of bone marrows of mice transplanted with BRAFV600E expressing HSPCs we detected senescence in both oncogene and non-oncogene expressing cells.”

4. Is the lethal histiocytic disease transplantable? If so, what does it take – HSPC isolated from diseased bone marrow, or just peripheral blood or cells from p16-high tissue lesions?

The Reviewer raises an interesting point on the transplantability of the histiocytic disease. Our data provide evidence that transplantation of BRAF^{V600E} expressing human HSPCs causes a lethal form of histiocytosis, in agreement with previous studies that show that mutated bone marrow-derived HSPCs from patients affected by Langerhans Cell Histiocytosis (LCH) and Erdheim-Chester Disease (ECD) can be transplanted into NSG mice, although with variable outcomes in terms of engraftment and reproducibility of the disease (PMID: 28566492). Thus, our work further corroborates the knowledge that HSPCs are the cell of origin of aggressive forms of histiocytosis and can engraft into primary recipient mice after transplantation. We also attempted at transplanting BRAF^{V600E}-expressing HSPCs into secondary recipients but unfortunately this set of experiments did not provide encouraging results as in our model (which displays a strong bone marrow failure phenotype with early onset) the BRAF^{V600E} -expressing human HSPCs are extremely rare (already at 2 weeks post transplantation) (see also New Figure 2K) and fully committed towards the myeloid lineage, rendering very challenging the experiments on transplantability in secondary mice. Regarding the possibility to transplant terminally differentiated p16-positive myeloid cells from histiocytic lesions, we found this idea fascinating but reasoned that these experiments would be time-sensitive and will require additional optimization of the protocol for isolation of alive senescent cells from the lesions. Moreover, because of the senescence program, those non-proliferating histiocytes will most likely not engraft into secondary recipient at significant levels, rendering necessary a scale-up of the primary transplant experiments to maximize input material for secondary transplants. Therefore, we would like to test this intriguing hypothesis in a follow-up study.

Minor concerns

1. Fig. 1F: It would have been more informative to see cells isolated from BRAF-V600E recipients being GFP+ and marker-positive (e.g. CD68, CD207, S100) by flow cytometry or immunofluorescence

Following this Reviewer's suggestion we analyzed additional serial lung sections by immunohistochemistry to detect GFP for vector marking, p16 as senescence marker, S100, CD207 and PGM1 as dendritic and monocytic markers and found that within the histiocytic lesions oncogene expressing (GFP⁺) cells co-express the aforementioned markers, confirming that the reported myeloid phenotype is associated to BRAF^{V600E} expression (New Figure 4I).

2. Fig. 2: The control group is a mixed bag of untransduced, GFP-only and BRAF-wt-infected HSPC – did it matter, whether the cells were actually infected (i.e. potential myeloid bias due to preferred vector integration sites)?

As reported in the Principal Component Analysis of our transcriptomic data (New Figure 3A) we did not observe major differences between the expression profile of myeloid cells expressing wtBRAF, only GFP or untransduced cells (to clarify we have now labeled individual samples in figure panel with the respective legend). Moreover, when we analyzed the impact of oncogene expressing on BM cellularity *in vivo* we did not detect any difference between the group of mice transplanted with untransduced cells or wtBRAF (New data in New Figure 1F and in New Supplementary Figures 1A-C and 2C, D). Therefore, unless otherwise specified, for analyses on hematopoietic reconstitution, myeloid skewing and expression profile *in vivo* we considered wtBRAF, only GFP or untransduced HSPCs as a unified control group. Finally, regarding the potential myeloid bias induced by vector integration, it is worth to point out that the probability of random insertional activation of cancer genes is very low, especially when lentiviral vectors with self-inactivating (SIN) LTRs and a moderate promoter in internal position are used (PMID: 16732270; PMID: 19307726), further supporting the assumption that the observed phenotype is essentially caused by the expression of the transgene, rather than random insertional mutagenesis or a general effect of vector integration alone.

Suppl. Fig. 2D indicates that there is a clear difference between BRAF-wt vs. BRAF-V600E, however, at an extremely low relative extent of CD19+ B-cells (accounting for around 2% of the overall cellularity). Is the relative loss of both BRAF-wt- and BRAF-V600E-positive CD19+ B-cells (Fig. 2O) due to OIS, also in response to wild-type BRAF overexpression?

This Reviewer is correct in pointing out that B-cells account only for the 2% of the total human graft (as shown in New Figure 2F and New Supplementary Figure 2F). However, the reported low percentage is related to the early time point of analysis (14 days after transplantation), when B-cells are still expanding in transplanted NSG mice and reach up to 60% of the human graft only around at least 15 weeks post transplant (see among others' publications PMID: 30905619; PMID: 28330619). Thus, we believe that the relatively small percent of B-cells in mice transplanted with wtBRAF expressing human HSPCs can be explained by the early timepoint of the analysis, dictated by the short lifespan of BRAF^{V600E} transplanted mice. Moreover, we would like to point out that while at 14 days after transplantation the CD19⁺ B cells in the wtBRAF group can still be detected, in the mice transplanted with the BRAF^{V600E}-expressing human HSPCs they are absent (New Figure 2J and Supplementary Figure 2G), indicating a robust cell-autonomous effect of oncogene activation on B cell lineage. We are afraid that the Reviewer mistakenly referred to Figure 2O when

discussing the B cell output because this figure panel is related to the Common Myeloid Progenitors (CMP) and not to B-cells.

3. Fig. 2J is not displayed properly in my PDF

We apologize for the issue with this figure panel. We now made sure that the item is displayed correctly.

Reviewer #2 (Remarks to the Author); expert on hematopoiesis and inflammation:

This manuscript entitled “Oncogene-induced senescence in hematopoietic progenitors features myeloid restricted hematopoiesis, chronic inflammation and histiocytosis” reports that oncogene BRAF V600E induces a senescence-like phenotype in human hematopoietic progenitors contributes to the bone marrow aplasia and histiocytosis development in a humanized mouse model. This manuscript also reports that the SASP factors (senescence-associated secretory phenotype) secreted by oncogene-induced senescent cells also induce a senescence-like phenotype in non-oncogene expressing cells, therefore triggering a cascade of systematic inflammation. This study provides valuable information to understand how small numbers of oncogene-induced senescent cells can initiate a vicious cycle and contribute to the deterioration of health.

Altogether this is an interesting manuscript that is in general well done. To assist the authors in substantiating their conclusions we have provided a list of concerns that need to be addressed.

We thank the Reviewer for his supportive evaluation of our study.

Major points:

1. The identity of the senescent cells from the bone marrow images in Figure 4 is not clear. p16 and beta-galactosidase can be induced in macrophages as part of a reversible response to physiological stimuli (PMCID 5611982) hence the result of these two markers in hematopoietic cells needs to be cautiously interpreted. The authors need to show whether these markers overlap with macrophages entirely or are also observed in non-macrophage human hematopoietic cells.

Likewise, the identity of the cells as human or mouse derived, V700E or WT is not clear and should be clarified either by imaging or flow cytometry to better support a model of in-trans induction of senescence.

We thank the Reviewer for the comments. We are aware of studies reporting p16 and SA- β -Gal accumulation in macrophages as part of the reversible response to inflammatory stimuli (discussed also in our recent review PMID: 33328614). To better understand if senescent markers are co-expressed with monocytic/macrophages markers, we performed new analyses on serial sections of lung tissues to characterize the histiocytic infiltrates by hematoxylin/eosin and immunohistochemistry, in order to detect GFP, and p16, S100, CD207 and PGM1 markers. Of note, the antibodies used for this study are specific for human antigens. The pathological analysis of the infiltrates confirmed that the morphological phenotype of the histiocytes within the lesions was heterogeneous and resembled those observed in human mixed histiocytosis, confirming that the reported myeloid phenotype is associated to the expression of BRAF^{V600E} and providing a more direct evidence of macrophage infiltration and senescence establishment of human origin (New Figure 4I). From this analysis, we also found that GFP and p16 were expressed together at high frequency and co-expressed the aforementioned monocytic/macrophages markers. However, not all histiocytes within the lesions were vector marked nor p16 positive and conversely several GFP-cells were p16 positive, indicating that also non-macrophage human hematopoietic cells express p16.

We have added these new results to the main text:

Page 11: “To better characterize senescence establishment in the histiocytic infiltrates, we performed immunohistochemical analysis of serial sections of lung lesions in mice transplanted with BRAF^{V600E}-expressing HSPCs, detecting vector marked GFP+ cells, p16+ cells, and the myeloid lineage markers S100, CD207 and PGM1. From this analysis we found that GFP and p16 were expressed together at high frequency and co-expressed different myeloid markers. However, not all histiocytes within the lesions were vector marked nor p16 positive and conversely several GFP- cells were p16 positive. The pathological analysis of the infiltrates confirmed that the morphological phenotype of the histiocytes within the lesions was heterogeneous and resembled those observed in human mixed histiocytosis (Figure 4I).”

It must be pointed out that we also assessed the activation of the DNA damage response (DDR) in the human graft by immunofluorescence analyses for the phosphorylated form of the apical DDR kinase ATM (New Supplementary Figure 4D), further supporting the idea that the observed increase in senescence markers is more likely due to the establishment of OIS through the accumulation of DNA damage, rather than to the transient activation of senescence markers to which the Reviewer is referring. To better corroborate the idea of senescence establishment in human myeloid cells upon oncogene activation, we have now also evaluated both p16 and SA-β-Gal on BM isolated from transplanted mice (New Figure 4K, L); of note, a hCD33 antibody was used and cells were gated within the hCD45+ population, confirming that the detected p16+ and SA-β-gal+ cells are indeed of human origin. In this new set of experiments, we also analyzed the expression of the aforementioned markers within the GFP+ and GFP- fraction. While we did not detect p16+ cells in the wtBRAF group, neither in the GFP- nor in the GFP+ fraction, we did observe that approximately 30% of BRAF^{V600E} cells (GFP+) express p16 and are positive for SA-β-gal. Interestingly, we also observe p16 expression within the GFP- fraction of the BRAF^{V600E} group, further supporting the idea of a strong paracrine effect (mediated by SASP) on bystander cells.

We have added the results of this analysis in the main text

Page 11: “FACS analysis showed that myeloid cells (CD33+) from the bone marrow of mice transplanted with BRAF^{V600E}-expressing HSPCs were significantly enriched for p16+ and SA-βGal+ cells compared to controls. This significant increase in senescent cells was also observed in the GFP- fraction, further supporting the establishment of senescence in bystander cells (Figure 4K, L and Supplementary Figure 4M).”

Along these lines, presumably senescence of progenitor cells is the trigger of the inflammatory phenotype based on the model proposed by the authors – to rigorously substantiate this, association of senescence markers with phenotypic HSPC *in vivo* should be provided.

The Reviewer raises an interesting but difficult-to-address point for which we cannot provide an answer timely. The Reviewer should consider that while we are confident of the establishment of senescence in BRAF^{V600E} expressing HSPCs cultured *in vitro* and in a substantial number of myeloid progenitors and fully differentiated histiocytes, it would be technically challenging to dissect senescence emergence in HSPC subsets *in vivo*, especially when considering also the low cellularity observed in BM obtained from transplanted mice from the BRAF^{V600E} group (New Figure 1F and New Supplementary Figure 1A, B). Furthermore, as we show in New Figure 2 K-P and Supplementary Figure 2L-M, the amount of BRAF^{V600E}-expressing (GFP⁺) within all human HSPCs subsets such as HSC (hematopoietic stem cells), MLP (multi-lymphoid progenitors), PreBNK (B and NK cell progenitors), CMP (Common-Myeloid Progenitors), GMP (Granulocyte/Monocyte Progenitors), CD90+ (HSC-enriched fraction), MPP (Multi-potent

progenitors), EP (Erythroid progenitors), MKP (Megakaryocyte Progenitors), MEP (Megakaryocyte/Erythrocyte Progenitors), ETP (Early T cell progenitors) is extremely low already at 14 days post transplant and it is dramatically affected by oncogene activation. Therefore, the suggested experiments of senescence evaluation in HSPC subsets *in vivo* would require analyzing a limited fraction of GFP⁺ cells within an already rare population of cells and to choose the best time point, which will change on the specific emergence of the HSPC subsets of interest upon transplant. Additional hurdles include the setup and the optimization of intracellular stainings for senescence markers (p16 and SA-β-Gal) in combination with multiple surface markers to identify different progenitor populations, which will require a sizable fraction of cells. In summary, we think that this analysis would be exceedingly difficult (from both biological and technical standpoints) to perform in a reasonable amount of time.

2. This manuscript also showed that sporadic p16⁺ cells were detected in lung sections. However, as in point 1) it is unclear whether these cells were human CD45 cells or mouse CD45 cells, or mouse non-hematopoietic cells present in the tissue. Defining the origin species of these cells will strengthen this manuscript's central claim.

We thank the Reviewer for this criticism. We would like to highlight that all antibodies used for senescence detection (p16) and for myeloid markers (CD14, CD1a, S100, CD207 and PGM1) are raised against epitopes of human origin. However, in order to define the impact of oncogene-activation on human or mouse cells, a point raised also by Reviewer 1, we have performed a new set of experiments aimed to comprehensively address the impact of transplantation of BRAF^{V600E} expressing human HSPC on mouse hematopoiesis. Mice were transplanted either with BRAF^{V600E} or wtBRAF expressing human HSPCs or with untreated HSPCs. After 24 days post transplantation, we reported that in the BRAF^{V600E} group total BM cellularity (measured by absolute cells number) was much lower than in mice from wtBRAF or untreated groups (New Figure 1F). The reduction in BM cellularity in BRAF^{V600E} group involved both murine and human CD45⁺ hematopoietic cells (New Supplementary Figure 1B, C), with a marked skewing of the murine hematopoiesis towards the monocytic/macrophagic lineage (CD11^{high}/Gr⁻/mCD45⁺) (New Supplementary Figure 2C), while the granulocytic compartment (CD11^{high}/Gr⁺/mCD45⁺) did not appear to be affected in any of the groups analyzed (New Supplementary Figure 2D). In summary, both human and mouse hematopoietic cells in the bone marrow appear to be strongly reduced upon transplantation of BRAF^{V600E} expressing human HSPCs *in vivo*, with a marked myeloid skewing involving also murine HSPCs, confirming the idea of a paracrine role of BRAF^{V600E}- expressing HSPCs not only on human bystander cells but also on murine HSPCs. To characterize the induction of OIS in HSPCs, we have performed new *in vivo* analyses to investigate p16 and senescence induction. FACS analyses on human CD33⁺ cells isolated from the BM of transplanted mice revealed that the percentage of p16⁺ cells was as high as 33% within the vector marked HSPCs from the BRAF^{V600E} group and of approximately 16% in bystander GFP⁻ cells (New Figure 4K), confirming that human cells within BM do express p16. Of note, for technical reasons we had to restrict our analyses on CD33⁺ cells, discarding CD19⁺ cells (which at the time of the analysis represent a low percentage of human cells within the graft).

Although it is likely that p16 may also be induced in murine cells through SASP factors produced by senescent human HSPCs, considering the high percentage of p16⁺ positive cells within the human graft and the fact that the antibody used for the analyses was raised against the human protein, we are confident that the cells highlighted as p16⁺ are of human origin. Even in the unfortunate event of the antibody recognizing murine p16, we would like to highlight that considering the fast kinetics of disease onset (within 24 days from transplantation), and that it takes at least 8 days post transduction (*in vitro*) to establish OIS, we can speculate that, *in vivo*, murine cells would start to undergo senescence much later than their human counterpart, and that mice would succumb to disease by the time we could detect a significant percentage of murine p16⁺

cells. In any case, dissection of the effect of oncogene-induced senescence in HSPCs of murine origin is in our interest and it is currently being investigated by our group also taking advantage of the *Cdkn2a*^{-/-} mouse model. Hopefully, we will be able to provide more experimental evidence on this aspect in a follow-up study in the near future.

3. Histiocytosis is defined as an overabundance of tissue macrophages leading to pathology. The authors should provide more direct evidence for macrophage infiltration in their human and mouse tissue sections.

As explained in response to point 1, we have now included additional tissue sections showing the histiocytic infiltrates by Hematoxylin/eosin and staining with antibodies against human p16 and monocyte/macrophage markers to provide more direct evidence of macrophage and dendritic cell infiltration in our mouse model (New Figure 4I) and in lesions from patients affected by Erdheim Chester Disease (New Supplementary Figure 4N).

Minor points:

1. The authors performed transduction efficiency analysis and they might already have the data of transduction efficiency in different populations, HSC, MLP, PreBNK, CMP, GMP. These data should be shown as FACS plots for each population to allow the reader to gauge transduction efficiency and GFP expression levels. Adding this information will help to rule out the possibility that impaired lymphoid lineage is not caused by biased transduction efficiency in MLP and/or CMP populations.

We now provide the FACS plots of the GFP levels of the different HSPC subsets analyzed (New Supplementary Figure 2Q). From this analysis we show that the MLP (lymphoid progenitor) subset has lower transduction levels compared to myeloid (CMP) subsets (New Supplementary Figure 2K). This slight difference does not justify the concern that the different transduction levels of myeloid or lymphoid progenitors could impact on the reported lymphoid impairment of the BRAF^{V600E} group. Indeed, being the lymphoid progenitors less transduced than myeloid progenitors we would expect myeloid cells to be more negatively affected by the oncogenic activation, which is not the case. We hope that these additional data clarify the Reviewer's concern.

We now added this consideration to our discussion:

Page 16: "Differences in transduction levels could not explain the observed phenotype as lymphoid progenitors had less vector marking than myeloid counterparts."

2. It is stated in the manuscript that "Macroscopic analysis of the internal organs at euthanasia showed markedly paler long bones in mice from the BRAFV600E group, suggesting a profound alteration of BM cellularity, accompanied by a modest spleen enlargement (Figure 1E)". A quantification for total BM cellularity will be needed to support this statement beyond the image of paler bones shown.

We thank the Reviewer for this comment. We now provide an absolute quantification of the numbers of total, human and mouse CD45⁺ cells within the BM and report significant BM aplasia in the BRAF^{V600E} group (New Figure 1F) involving both human and mouse CD45⁺ cells (New Supplementary Figure 1A, B).

We have added the results of this analysis in the main text

Page 5: "The overall BM cellularity of mice transplanted with BRAF^{V600E} transduced HSPCs was significantly reduced when compared to mice transplanted with untransduced or wtBRAF

expressing HSPCs (Figure 1F). This reduction in BM cellularity in the mice of the BRAF^{V600E} group involved both murine and human CD45+ cells (Supplementary Figure 1A, B) and was associated to a modest, although significant, increase of necrotic cells compared to controls (Supplementary Figure 1C)."

3. BRAF V600E induced necrosis was not ruled in or out in this study. Multiple studies have shown that BRAF mutations cause ulceration in melanoma patients (e.g., PMC4984864), an indication of tissue necrosis. Necrotic tissue triggers inflammation. Whether BRAF V600E overexpression in CD34+ cells induces necrosis is not addressed in this manuscript. Therefore, to draw a conclusion of oncogene induced senescence is the major driver for the phenotype described in this manuscript, the authors need to perform apoptosis and/or necrosis analysis on human CD34+ cells post BRAF V600E lentivirus transduction. Indeed, cell necrosis could also explain as an alternative hypothesis to TNF the trans-acting inflammatory phenotype.

We thank the reviewer for pointing out this important aspect. We characterized necrotic cells in the BM of transplanted mice of different experimental groups (wtBRAF-BRAF^{V600E}-untransduced) and reported a very modest although significant increase in necrotic cells in the BM of mice from the BRAF^{V600E} group (Supplementary Figure 1C). We have complemented these data by performing an in depth characterization of apoptosis in BRAF^{V600E} or wtBRAF expressing HSPCs by annexinV and 7AAD staining to detect alive, early apoptotic and necrotic HSPCs. From this analysis we show that the majority of the cells were alive (95%) with only a limited number of apoptotic or necrotic cells. Of note, we did not detect any difference between control or BRAF^{V600E}-expressing HSPCs (New Supplementary Figure 4K), further highlighting that senescence rather than apoptosis is the major determinant of the detrimental impact of oncogene activation on human hematopoiesis.

Reviewers' comments:

Reviewer #1 (Remarks to the Author):

Biavasco-R,... ..Montini-E, Oncogene-induced senescence in hematopoietic progenitors features myeloid restricted hematopoiesis, chronic inflammation and histiocytosis
Revised version re-submitted to Nature Communications

This is now the revised version of the manuscript.

I have to acknowledge the authors' ability to respond to my comments and concerns carefully, to address them partly (and where technically feasible in reasonable time) experimentally, partly in re-phrasing or clarifying the respective sections in the manuscript – thus, in summary, satisfyingly at all crucial points. As a result, this interesting manuscript is now substantially improved, and I have no further objection against its publication in Nature Communications.

Only two minor points remain: (1) Fig. 4M lacks quantification. And (2): It is not clear from text and methods what the newly introduced sh-p16 lentiviral vector actually targets – p16INK4a, or both p16INK4a and p19ARF. This should be clarified in the methods, and for better comprehensibility, in the respective text passage as well.

Reviewer #2 (Remarks to the Author):

The authors have addressed my key concerns.

Response to Reviewers

Reviewer #1 (Remarks to the Author):

Biavasco-R,... ...Montini-E, Oncogene-induced senescence in hematopoietic progenitors features myeloid restricted hematopoiesis, chronic inflammation and histiocytosis
Revised version re-submitted to Nature Communications

This is now the revised version of the manuscript.

I have to acknowledge the authors' ability to respond to my comments and concerns carefully, to address them partly (and where technically feasible in reasonable time) experimentally, partly in re-phrasing or clarifying the respective sections in the manuscript – thus, in summary, satisfyingly at all crucial points. As a result, this interesting manuscript is now substantially improved, and I have no further objection against its publication in Nature Communications.

We thank again the Reviewer 1 for the appreciation of our study.

Only two minor points remain:
(1) Fig. 4M lacks quantification.

We thank the Reviewer for noticing this issue, we have added the quantification requested. Of note, by editor's request, we had to split figure 4 in 2 Figures, therefore previous Figure 4m is now the new Figure 5e.

And (2): It is not clear from text and methods what the newly introduced sh-p16 lentiviral vector actually targets – p16INK4a, or both p16INK4a and p19ARF. This should be clarified in the methods, and for better comprehensibility, in the respective text passage as well.

We thank the Reviewer for highlighting this matter. Our sh-p16 lentiviral vector does actually target both p16INK4a and p19ARF locus. We have updated both Methods and Results sections accordingly.

Reviewer #2 (Remarks to the Author):

The authors have addressed my key concerns.

We thank again the Reviewer 2 for the appreciation of the revisions we applied to our study.